# CompanyKG: A Large-Scale Heterogeneous Graph for Company Similarity Quantification

## Abstract

In the investment industry, it is often essential to carry out fine-grained company similarity quantification for a range of purposes, including market mapping, competitor analysis, and mergers and acquisitions. We propose and publish a knowledge graph, named CompanyKG, to represent and learn diverse company features and relations. Specifically, 1.17 million companies are represented as nodes enriched with company description embeddings; and 15 different inter-company relations result in 51.06 million weighted edges. To enable a comprehensive assessment of methods for company similarity quantification, we have devised and compiled three evaluation tasks with annotated test sets: similarity prediction, competitor retrieval and similarity ranking. We present extensive benchmarking results for 11 reproducible predictive methods categorized into three groups: node-only, edge-only, and node+edge. To the best of our knowledge, CompanyKG is the first large-scale heterogeneous graph dataset originating from a real-world investment platform, tailored for quantifying inter-company similarity.

## 1 Introduction

In the investment industry, it is often essential to identify similar companies for purposes such as **market mapping**, **competitor analysis** and **mergers and acquisitions (M&A)**[1]. The significance of similar and comparable companies in these endeavors is paramount, as they contribute to informing investment decisions, identifying potential synergies, and revealing areas for growth and improvement. Therefore, the accurate quantification of inter-company similarity (a.k.a., company similarity quantification) is the cornerstone to successfully executing such tasks. Formally, **company similarity quantification refers to the systematic process of measuring and expressing the degree of likeness or resemblance between two companies with a focus on various "traits", typically represented through numerical values**. While a universally accepted set of "traits" does not exist, investment researchers and practitioners commonly rely on attributes, characteristics or relationships that reflect (but not limited to) **competitive landscape**, **industry sector**, **M&A transactions**, **people's affiliation**, **news/event engagement**, and **product positioning**.

In real investment applications, company similarity quantification often forms a central component in a **similar company expansion** system, which aims to extend a given set of companies by incorporating the similarity traits mentioned above[2]. Such a system aids in **competitor analysis**, where investment professionals begin with a **seed company** $x_0$ of interest (a.k.a. **query company**, represented by a company ID), aiming to identify a concise list of direct competitors to $x_0$. The identified competitor companies serve as benchmarks for evaluating the investment potential of $x_0$, influencing the final investment decisions. Until now, implementing such a company expansion system remains a challenge, because (1) the crucial business impact necessitates an exceptionally high production-ready standard of recommendation recall and precision; and (2) the underlying data has expanded to an unprecedented scale and complexity, particularly regarding the intricate and diversified inter-company relationships.

In recent years, transformer-based Language Models (LMs), such as Cer et al. (2018); Devlin et al. (2019); Reimers & Gurevych (2019), have become the preferred method for encoding textual company descriptions into vector-space embeddings. Companies that are similar to the seed companies can be

---

[1] **Market mapping** analyzes the market landscape of a product or service, while **competitor analysis** focuses on the competitive landscape of a concrete company (Zeisberger et al., 2017). They both start with identifying a set of similar companies, yet the former often expects a more coarse and inclusive result than the latter. **M&A** refers to the process of combining two or more companies to achieve strategic and financial objectives, such as enhanced market share and profitability (Sherman, 2018). Investment professionals typically work with a list of M&A candidate companies and examine them based on the sequence of their projected business compatibility.

[2] In domains beyond investment, related research such as (Lissandrini et al., 2020; Akase et al., 2021) focuses on expanding query entities to a more extensive and relevant set, akin to our work. We emphasize the difference to a **company suggestion** system which concentrates on proactively recommending entities based on user behavior and preferences. However, company expansion and suggestion are both concepts in the context of KG and **recommender systems** with slightly different processes and implication.

searched in the embedding space using distance metrics like cosine similarity. Rapid advancements in Large LMs (LLMs), such as GPT-3/4 (Brown et al., 2020; OpenAI, 2023) and LLaMA (Touvron et al., 2023), have significantly enhanced the performance of general-purpose conversational models. Models such as ChatGPT (Ouyang et al., 2022) can, for example, be employed to answer questions related to similar company discovery and quantification in an $N$-shot prompting paradigm, where $N \geq 0$ examples are included in the input prompt to guide the output of a language model.

A graph is a natural choice for representing and learning diverse company relations due to its ability to model complex relationships between a large number of entities. By representing companies as nodes and their relationships as edges, we can form a **knowledge graph** (KG)[3]. A KG allows us to efficiently capture and analyze the network structure of the business landscape. Moreover, KG-based approaches allow us to leverage powerful tools from network science, graph theory, and graph-based machine learning, such as Graph Neural Networks (GNNs) (e.g., Hamilton et al., 2017a; Veličković et al., 2019; Zhu et al., 2020; Hassani & Khasahmadi, 2020; Hou et al., 2022; Tan et al., 2023), to extract insights and patterns to facilitate similar company analysis that goes beyond the relations encoded in the graph. While there exist various company datasets (mostly commercial/proprietary and non-relational) and graph datasets for other domains (mostly for single link/node/graph-level predictions), there are to our knowledge no datasets and benchmarks that target learning over a large-scale KG expressing rich pairwise company relations.

We gain access to a general-purpose **investment platform**[4] that integrates over 50 data sources about companies, providing investment professionals (i.e., users) with a comprehensive and up-to-date view of the companies and the entire market. Meanwhile, valuable insights are collected from the interaction between the users and the platform, which often implies inter-company relations of various kinds. Based on this unique edge, we construct and publish a large-scale heterogeneous graph dataset to benchmark methods for similar company quantification. Our main contributions include:

- We introduce a real-world KG dataset for similar company quantification – **CompanyKG**[5]. Specifically, 1.17 million companies are represented as nodes enriched with company description embeddings; and 15 distinct inter-company relations result in 51.06 million weighted edges.
- To facilitate a comprehensive assessment of methods learned on CompanyKG, we design and collate three evaluation tasks with annotated datasets: **Similarity Prediction** (**SP**), **Competitor Retrieval** (**CR**) and **Similarity Ranking** (**SR**).
- We provide comprehensive benchmarking results (with source code for reproduction) of **node-only** (embedding proximity and $N$-shot prompting), **edge-only** (shortest path and direct neighbors) and **node+edge** (self-supervised graph learning) methods.

## 2 RELATED WORK

Typical efforts to define company similarity employ a multidimensional approach, encompassing various facets such as industry classifications (Hoberg & Phillips, 2010), financial flexibility (Hoberg et al., 2014) and shared directorship (Lee et al., 2021). Among the widely embraced data sources tailored for the investment domain – most of which are not in the format of a KG – are Diffbot, Pitchbook, Crunchbase, CB Insights, Tracxn, and S&P Capital IQ[6]. They primarily offer predicted similar companies without revealing the methodology or foundational data underpinning the predictions. In recent years, investment practitioners seeking a scalable method for recommending companies similar to seed entities have predominantly relied on **LMs** or **KGs**. Our benchmark tasks and methods focuses primarily on graph-based company similarity measurement for a task like competitor analysis. However, a company KG of this sort could be used for other tasks, where other modeling and querying approaches would be appropriate (e.g., Jayaram et al., 2014; Lissandrini et al., 2020). Alternatively, non-KG-based approaches for entity similarity quantification can build on similar sources of company data, just as textual descriptions or keywords. We include two here (embedding proximity and LLM prompting) for comparison to KG-based approaches, although other options exist, such as those grounded in document co-occurrence (Foley et al., 2016) or the fusion of a KG with domain-specific signals (Akase et al., 2021).

---

[3]A knowledge graph is a specific type of graph where nodes represent real-world entities/concepts and edges denote relations between the entities.

[4]The details of the investment platform (also referred to as "**the platform**") are hidden for review.

[5]The dataset and source code (utility and benchmarking) will be public on Zenodo and GitHub respectively.

[6]Diffbot: `www.diffbot.com`; Pitchbook: `www.pitchbook.com`; Crunchbase: `www.crunchbase.com`; CB Insights: `www.cbinsights.com`; Tracxn: `www.tracxn.com`; S&P Capital IQ: `www.capitaliq.com`. These data sources can be used beyond similar company quantification.

### 2.1 LANGUAGE MODEL (LM) BASED APPROACHES

**Text embedding proximity** has become a popular approach for company similarity quantification. The key idea is to represent textual descriptions of companies as dense vectors (a.k.a., embeddings) in a high-dimensional space, such that the similarity between two companies can be measured by the proximity between their corresponding vectors. The text embeddings are usually obtained using pretrained LMs, such as BERT (Devlin et al., 2019), T5 (Raffel et al., 2020), LLaMA (Touvron et al., 2023) and GPT-3/4 (Brown et al., 2020; OpenAI, 2023). The pretrained LMs can be used directly, or finetuned on company descriptions in a supervised (Cer et al., 2018; Reimers & Gurevych, 2019), semi-supervised (Cao et al., 2021), or Self-Supervised Learning (SSL) paradigm (Gao et al., 2021) to further improve the performance of proximity search. Supervised and semi-supervised methods typically yield superior performance (than SSL) on domain-specific tasks, when high-quality annotations are available (Cao et al., 2021). This limits its applicability in scenarios where the annotation is scarce, noisy, or costly to obtain.

$N$-**shot prompting**. The recent rapid development of LLMs such as GPT-3/4 (Brown et al., 2020; OpenAI, 2023), LLaMA (Touvron et al., 2023) and LaMDA (Thoppilan et al., 2022) has led to significant improvements in general-purpose conversational AI like ChatGPT (Ouyang et al., 2022). By prompting them with $N \geq 0$ examples, the instruction tuned models can answer questions about identifying similar companies, for example "*Can you name 10 companies that are similar to OpenAI?*" As a result, $N$-shot prompting has emerged as a potential tool for investment professionals (e.g., Wu et al., 2023; Yue & Au, 2023) looking to conduct similar company search and analysis. However, this approach is currently limited by several factors: (1) to ensure the model's responses are up-to-date, relevant, and precise, a large amount of domain-specific information must be incorporated, an active area of research with various methods such as contextual priming and fine-tuning being explored; (2) the decision-making process may not be explainable, making it difficult to understand and trust the answer; (3) the performance is closely tied to meticulous design of prompts.

### 2.2 KNOWLEDGE GRAPH (KG) BASED APPROACHES

Companies can be represented as nodes in a graph, where each node is enriched with attributes. There can be various types of similarities between any two companies, which can be naturally represented as heterogeneous edges in the graph. Moreover, the strength of each relation can be captured by assigning appropriate edge weights. Such a graph-based representation enables the use of heuristic algorithms or GNN models for searching and identifying similar companies, based on their structural and attribute-level similarities. According to our survey[7], most public graph datasets are designed for predicting node (Zeng et al., 2019), edge (Bollacker et al., 2008) or graph (Rozemberczki et al., 2020) level properties, while our graph is tailored for company similarity quantification[8]. The most common entities in KGs are web pages (Mernyei & Cangea, 2020), papers (Bojchevski & Günnemann, 2018), particles (Brown et al., 2019), persons (Leskovec & Mcauley, 2012) and so on. Relato Business Graph[9], which also represents companies as node entities, is the dataset most similar to ours. However, its limited scale and absence of edges with explicit implications of company similarity could constrain its suitability for quantifying company similarity. To address this, we build CompanyKG, a large-scale company KG that incorporates diverse and weighted similarity relationships in its edges. We present extensive benchmarking results for 11 reproducible baselines categorized into three groups: node-only, edge-only, and node+edge. To the best of our knowledge, CompanyKG is the first large-scale heterogeneous graph dataset originating from a real-world investment platform, tailored for quantifying inter-company similarity.

## 3 COMPANYKG

### 3.1 EDGES (RELATIONS): TYPES AND WEIGHTS

We model 15 different inter-company relations as undirected edges, each of which corresponds to a unique edge type (ET) numbered from 1 to 15 as shown in Table 1. These ETs capture six widely adopted categories (C1~C6) of similarity between connected company pairs: C1 - competitor

---

[7]We have reviewed graph datasets in SNAP (Leskovec & Krevl, 2014), OGB (Hu et al., 2020), Network Repository (Rossi & Ahmed, 2015), Hugging Face (https://huggingface.co), Kaggle (https://www.kaggle.com) and Data World (https://data.world)

[8]Formally, it is equivalent to edge prediction, yet edge prediction methods like GraphSAGE usually assume an exhaustive edge representation in the graph, which is not the case in CompanyKG as discussed in Section 3.1.

[9]https://data.world/datasyndrome/relato-business-graph-database

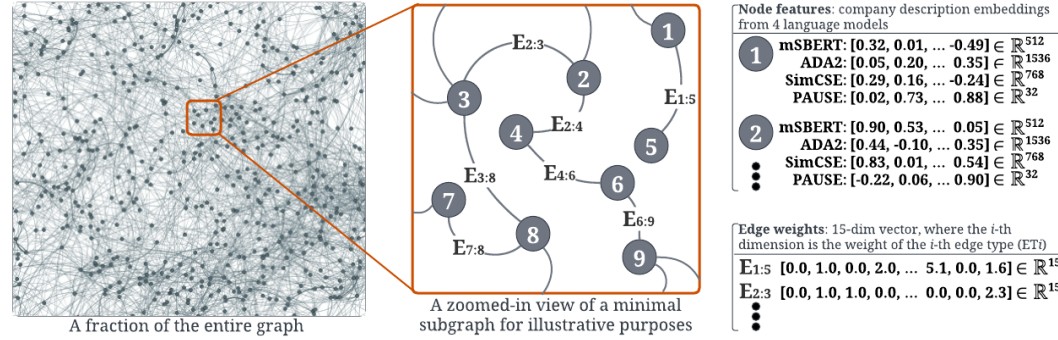

Figure 1: An illustrative subgraph (center) of the heterogeneous and undirected CompanyKG graph (left), where the numbered nodes represent distinct companies and the edges signify multi-dimensional inter-company relations. Example node features and edge weights are displayed on the right.

landscape (ET2, 5 and 6), C2 - industry sector (ET3, 4, 9 and 10), C3 - M&A transaction (ET1, 7 and 11), C4 - people's affiliation (ET8 and 15), C5 - news/events engagement (ET12 and 14), and C6 - product positioning (ET13). The constructed edges do not represent an exhaustive list of all possible edges due to incomplete information (more details in Appendix F.2.③). Consequently, this leads to a sparse and occasionally skewed distribution of edges for individual ETs. Such characteristics pose additional challenges for downstream learning tasks. Associated with each edge of a certain type, we calculate a real-numbered weight as an approximation of the similarity level of that type. The edge definition, distribution, and weights calculation are elaborated in Table 1. As depicted in Figure 1, we merge edges between identical company pairs into one, assigning a 15-dim weight vector. Here, the $i$-th dimension signifies the weight of the $i$-th ET. Importantly, a "0" in the edge weights vector indicates an "*unknown relation*" rather than "*no relation*", which also applies to the cases when there is no edge at all between two nodes. See Appendix C for distribution of edge weights. Although there are a few ETs that can naturally be treated as directed (e.g. ET7, M&As), most are inherently undirected. We therefore treat the whole graph as undirected.

## 3.2 Nodes (Companies): Features and Profile Analysis

The CompanyKG graph includes all companies connected by edges defined in the previous section, resulting in a total of 1,169,931 nodes. Each node represents a company and is associated with a descriptive text, such as "*Klarna is a fintech company that provides support for direct and post-purchase payments ...*". As mentioned in Appendix F.2, about 5% texts are in languages other than English. To comply with privacy and confidentiality requirements, the true company identities or any raw interpretable features of them can not be disclosed. Consequently, as demonstrated in the top-right part of Figure 1, we encode the text into numerical embeddings using four different pretrained text embedding models: **mSBERT** (multilingual Sentence BERT) (Reimers & Gurevych, 2019), **ADA2**[10], **SimCSE** (Gao et al., 2021) (fine-tuned on textual company descriptions) and **PAUSE** (Cao et al., 2021). The choice of embedding models encompasses both pre-trained, off-the-shelf (mSBERT and ADA2) and proprietary fine-tuned (SimCSE and PAUSE) models. Justifications and details of node features/embeddings can be found in Appendix F.2. To showcase the sophistication and inclusiveness of the company profiles, Appendix B presents the distribution of companies across five attributes, revealing a wide dispersion of companies.

## 3.3 Evaluation Tasks

The main objective of CompanyKG is to facilitate developing algorithms/models for recommending companies similar to given seed/query companies. The effectiveness of such recommendations relies on the capability of quantifying the degree of similarity between any pair of companies. Two-stage paradigms are prevalent in large-scale recommender systems (Wang & Joachims, 2023): the first stage swiftly generates candidates that fall into the similarity ballpark, while the second stage ranks them to produce the final recommendations. We assemble a **Competitor Retrieval (CR)** task from real investment deals to assess the end-to-end performance over the entire task of company expansion. To understand the performance for each stage, we carefully curate two additional evaluation tasks: **Similarity Prediction (SP)** and **Similarity Ranking (SR)**. We explain below how SP and SR address different real-world investment use cases.

---

[10]ADA2 is short for "text-embedding-ada-002", which is OpenAI's most recent embedding engine employing a 1536-dimensional semantic space, with default settings used throughout the experiments.

| Type | The specification of each edge type (ET) and the associated weight | #edge |
|---|---|---|
| ET1 (C3) | In the platform, investment professionals (i.e., users) can create a so-called "space" by specifying a theme such as "*online music streaming service*". In each space, users are exposed to some companies that are recommended (by a linear model) as theme-compliant. Users can approve or reject each recommended company, which will be used to continuously train the model in an active learning fashion. We create an edge between any two confirmed companies in the same space. For any two companies with an edge with this type, the associated edge weight counts their approval occurrences in all spaces. | 117,924 |
| ET2 (C1) | The users of the investment platform can add $N \geq 1$ direct competitors for any target company, resulting in a competitor set containing $N+1$ companies ($N$ direct competitors and the target company itself). For each target company with such a competitor set, we create one fully connected sub-graph. When merging these sub-graphs, the duplicated edges are merged into one with a weight indicating the degree of duplication. | 16,734 |
| ET3 (C2) | Investment professionals can create "collections" of companies for various purposes such as market mapping and portfolio tracking. We manually selected 787 (out of 1458 up till February 2023) collections that contain companies with a certain level of similarity. We create a fully connected sub-graph for each collection. These sub-graphs are then merged and deduplicated like **ET2**, so the edge weights represent the duplication degree. | 374,519 |
| ET4 (C2) | Based on a third-party data source* containing about 4 million company profiles, we create graph edges from company market connections. The edge weights represent the count of the corresponding market connection in that data source. | 1,473,745 |
| ET5 (C1) | From a third-party data source* about companies, investors and funding rounds, we extract pairwise competitor relations and build edges out of them. Again, the edge weights are the number of occurrences of such extracted relation. | 74,142 |
| ET6 (C1) | Same as **ET5** except that competitor information is extracted from a different data source* that specializes in financials of private and public companies. | 346,162 |
| ET7 (C3) | The platform also works with deal teams on various M&A and add-on acquisition† projects. Such projects aim to produce a list of candidate companies for a certain company to potentially acquire. For each selected project* up till December 2022, we create a fully connected sub-graph from a company set containing the acquirer and all acquiree candidates. These sub-graphs are then merged and deduplicated like **ET2** and **ET3**, so the edge weights represent the duplication degree. | 55,366 |
| ET8 (C4) | Based on the naïve assumption that employees tend to move among companies that have some common traits, we build graph edges representing how many employees have flowed (since 1990) between any two companies. For any two companies A and B who currently have $N_A$ and $N_B$ employees respectively, let $N_{A:B}$ and $N_{B:A}$ respectively denote the number of employees flowed from A to B and from B to A. We create an edge between A and B either when (1) $N_{A:B} + N_{B:A} \geq 20$, or (2) $N_{A:B} + N_{B:A} \geq 3$ and $\frac{N_{A:B}+N_{B:A}}{N_A+N_B} \geq 0.15$. We assign the value of $\frac{N_{A:B}+N_{B:A}}{N_A+N_B}$ as the edge weight. | 71,934 |
| ET9 (C2) | In the platform's data warehouse, each company has some keywords (e.g., company OpenAI may have three keywords: *artificial intelligence*, *machine learning* and *ChatGPT*) that are integrated from multiple data sources*. We filter out the overly short/long keywords and obtain 763,503 unique keywords. Then we calculate IDF (inverse document frequency) score for every keyword and apply min-max normalization. Finally, we create an edge between any two companies that have shared keyword(s), and assign the average (over all shared keywords) IDF score as the corresponding edge weight. | 45,717,108 |
| ET10 (C2) | The platform adopts a hierarchical sector framework to support thematic investment activities. The users of the platform can tag a company as belonging to one or more sectors (e.g., *cyber security*, *fintech*, etc.). For each low-level sector (i.e., sub-sector and sub-sub-sector), we create a fully connected sub-graph from all the tagged companies. The sub-graphs are further merged into one big graph with edge weights indicating duplication degree of edges. It also worth mentioning that an edge created from a sub-sub-sector is weighted twice as important as the one created from a sub-sector. | 813,585 |
| ET11 (C3) | Mergers and/or acquisitions could imply a certain level of similarity between the acquirer and acquiree. As a result, we create an edge between the involved companies of historical (since 1980) merger/acquisition (specifically M&A, buyout/LBO, merger-of-equals, acquihire)‡ events. The edge weight is the number of occurrences of the company pair in those events. | 260,644 |
| ET12 (C5) | Based on a third-party data source* that keeps track of what events (e.g., conferences) companies have attended, we create a preliminary edge between two companies who have had at least five co-attendance events in the past (up till June 2022). Then, we further filter the edges by computing the Levenshtein distance between the company specialties that come with the data source in the form of a textual strings. The edge weights are the count of co-attendance in log scale. | 1,079,304 |
| ET13 (C6) | From a product comparison service*, we obtain data about what products are chosen (by potential customers) to be compared. By mapping the compared products to the owning companies, we can infer which companies have been compared in the past up till February 2023. We build a graph edge between any two companies whose products has been compared. The edge weights are simply the number of times (in log scale) they are compared. | 216,291 |
| ET14 (C5) | The platform has a news feed* about global capital market. Intuitively, companies mentioned in the same piece of news might be very likely connected, thus we create an edge between any pair of the co-mentioned companies (till February 2023). When one of the co-mentioned companies is an investor, the relation is often funding rather than similarity, therefore we eliminate those edges from the final graph. The edge weights are log-scale co-mention count. | 151,302 |
| ET15 (C4) | One individual may hold executive roles in multiple companies, and it is possible for them to transition from one company to another. When any two companies, each having fewer than 1,000 employees, share/have shared the same person in their executive role, we establish a graph edge between them. To address the weak and noisy nature of this similarity signal, we refine the edges by only retaining company pairs that share at least one keyword (cf. **ET9**). The edge weights are log-scale count of shared executives between the associated companies. | 273,851 |

* The detailed information is hidden due to legal and compliance requirements. Our intent is not to re-license any third-party data; instead, we focus on sharing aggregated and anonymized data for pure research purposes only. Please refer to Appendix F.5 for more information about dataset usage.
† An add-on acquisition is a type of acquisition strategy where a company acquires another company to complement and enhance its existing operations or products.
‡ Leveraged buyout (LBO) is a transaction where a company is acquired with a significant loan. Acquihire is the process of acquiring a company primarily to recruit its employees, rather than to gain control of its products/services. A merger-of-equals is when two companies of about the same size come together to form a new company.

Table 1: Specification of 15 edge types (ET1∼ET15), edge weights (stronger relations has higher weights), and the number of edges per type ("#edge" column). ETs fall into six extensively recognized categories of company similarity: C1 (competitor landscape), C2 (industry sector), C3 (M&A transaction), C4 (people's affiliation), C5 (news/events engagement), and C6 (product positioning).

**Competitor Retrieval (CR)** describes how a team of experienced investment professionals would carry out **competitor analysis**, which will be a key application of a company expansion system. Concretely, investment professionals perform deep-dive investigation and analysis of a **target company** once they have identified it as a potential investment opportunity. A critical aspect of this process is competitor analysis, which involves identifying several **direct competitors** that share significant similarities (from perspectives of main business, marketing strategy, company size, etc.) with the target company. The investment platform maintains a proprietary archive of deep-dive documents that contain competitor analysis for many target companies. We select 76 such documents from the past 4

years pertaining to distinct target companies and ask human annotators to extract direct competitors from each document, resulting in ∼5.3 competitors per target on average. Ideally, for any target company, we expect a competent algorithm/model to retrieve all its annotated direct competitors and prioritize them at the top of the returned list.

**Similarity Prediction (SP)** defines the coarse binary similar-versus-dissimilar relation between companies. Excelling in this task often translates to enhanced recall in the first stage of a recommender system. It also tends to be valuable for use cases like **market mapping**, where an inclusive and complete result is sought after. For SP task, we construct an evaluation set comprising 3,219 pairs of companies that are labeled either as positive (similar, denoted by "1") or negative (dissimilar, denoted by "0"). Of these pairs, 1,522 are positive and 1,697 are negative. Positive company pairs are inferred from user interactions with the investment platform – closely related to ET2&3, although the pairs used in this test set are selected from interactions after the snapshot date, so we ensure there is no leakage. Every negative pair is formed by randomly sampling one company from a company collection (cf. ET3 in Table 1) and another company from a different collection. All samples, including the negative pairs, have been manually examined by domain experts.

**Similarity Ranking (SR)** is designed to assess the ability of any method to rank **candidate companies** (numbered 0 and 1) based on their similarity to a **query company**. Excelling in this task is of utmost significance for the second stage of a high-quality recommender system. When applied to real-world investment scenarios like **M&A**, a fine-grained ranking capability can significantly enhance the efficacy of prioritizing M&A candidates. During the creation of this evaluation dataset, we need to ensure a balanced distribution of industry sectors and increase the difficulty level of the task. Therefore, we select the query and candidate companies using a set of heuristics based on their sector information. As a result, both candidates will generally be quite closely related to the target, making this a challenging task. As described in Appendix F.3.④ and F.4.③, paid human annotators, with backgrounds in engineering, science, and investment, were tasked with determining which candidate company is more similar to the query company. Each question was assessed by at least three different annotators, with an additional annotator involved in cases of disagreement. The final ground-truth label for each question was determined by majority voting. This process resulted in an evaluation set with 1,856 rigorously labeled ranking questions. See Appendix F.3.② for more details. **We retained 20% (368 samples) as a validation SR set for model selection.** The choice to use the SR task for model selection, instead of the SP or CR tasks, is primarily because it covers a more representative sample of the entire graph. See Appendix F.2.⑧ for further explanation.

## 4 EXPERIMENTS

In addition to the CompanyKG dataset, we provide comprehensive benchmarks from 11 popular and/or state-of-the-art baselines categorized into three groups: **node-only**, **edge-only**, and **node+edge**.

**Node-only baselines** use only the features of individual companies provided in the graph and ignore the graph structure. We measure the cosine similarity between the embeddings of a given type (see Section 3.2) that are associated with any two companies in the graph. This gives us an **embedding proximity** score, which can be evaluated in the SP task, and a method of ranking candidates, evaluated by SR and CR. In order to assess the cutting-edge **N-shot prompting** methodology, we transform each raw SP/SR/CR evaluation sample into a text prompt and let ChatGPT (Ouyang et al., 2022) generate the label. The templates of task-specific prompts are provided in Appendix D.5.

**Edge-only baselines** merely utilize the graph structure. We define four simple heuristics to rank companies $C_i$ by proximity to a target company $T$: (1) **Unweighted Shortest Path** (**USP**) length from $T$ to $C_i$; (2) total weight of the **Weighted Shortest Path** (**WSP**) from $T$ to $C_i$ (weights inverted so lower is more similar); (3) close proximity only if $T$ and $C_i$ are immediate **Neighbors**; and (4) rank weighted neighbors (abbreviated as **W. Neighbors**) – immediate neighbors to $T$ ranked by edge weight. All four heuristics will choose randomly when the heuristic is not discriminative. To allow comparison of edge and path weights from different ETs, we scale edge weights to $[0, 1]$ for ET and shift them so they have a mean weight of 1. In order to avoid edge weights playing too strong a role in WSP compared to presence/absence of edges and paths, we add a constant[11] to all weights.

**Node+edge baselines** leverage both the node feature and graph structure. CompanyKG does not provide an explicit signal that can guide supervised learning to combine these into a single company

---

[11]We set this constant to 5 after a few empirical explorations. Results could potentially be improved by tuning this constant, but the method is intended here only as a simple baseline.

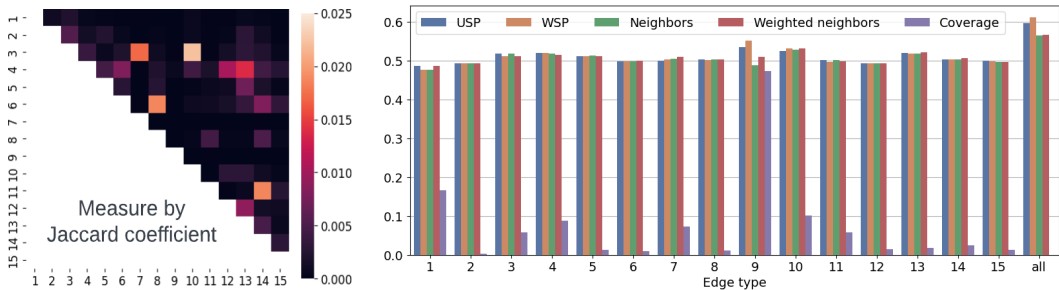

Figure 2: Pairwise redundancy analysis of ETs.

Figure 3: Ranking performance (SR test set) of different ETs using USP and WSP heuristics. Coverage for "all" is 100%.

representation for similarity measurement. As a result, we test five popular self-supervised GNN methods: **GRACE** (Zhu et al., 2020), **MVGRL** (Hassani & Khasahmadi, 2020),**GraphSAGE** (Hamilton et al., 2017b), **GraphMAE** (Hou et al., 2022) and **eGraphMAE** (Hou et al., 2022; Wang et al., 2021). GRACE and MVGRL are graph contrastive learning (GCL) approaches that construct multiple graph views via stochastic augmentations before learning representations by contrasting positive samples with negative ones (Zhu et al., 2021). GraphSAGE, which is especially suited for edge prediction, is trained to ensure that adjacent nodes have similar representations, while enforcing disparate nodes to be distinct. In light of the recent success of generative SSL in Natural Language Processing (NLP) and Computer Vision (CV) domains, Hou et al. proposed GraphMAE, a generative SSL method for graphs that emphasizes feature reconstruction. To incorporate the edge weights in GraphMAE, we replaced the graph attention network (GAT) (Veličković et al., 2018) with an *edge-featured* GAT (eGAT) (Wang et al., 2021), resulting in eGraphMAE (edge-featured GraphMAE) detailed in Appendix D.4. Unless otherwise specified, we report the best results from a hyper-parameter grid search detailed in Appendix D, using the validation split of the SR task set described in Section 3.3. We treat embedding type as a hyperparameter, optimized for each GNN method on the SR validation set, reporting only a single embedding type per method. However, results for all embedding types are given in Table 7 in the Appendix.

## 4.1 COMPARISON OF ETS (EDGE TYPES)

Before presenting the results of the evaluations on the full graph, we intend to gain some insights into the differences between the 15 different ETs. To understand the extent of redundancy among different ETs, we measure the overlap between any two ETs, such as ET$i$ and ET$j$. Letting $\{ETi\}$ and $\{ETj\}$ be the set of unweighted edges belonging to ETs $i$ and $j$ respectively, we measure the Jaccard coefficient $J = |\{ETi\} \cap \{ETj\}|/|\{ETi\} \cup \{ETj\}|$, where $|\cdot|$ denotes the size of the embodied set. The heatmap in Figure 2 shows that for most pairs of ETs, there is effectively no overlap, whilst for a few (e.g., 3–10, 11–14, 6–8) there is a slightly stronger overlap. However, the coefficient for the most overlapped pair, 3–10, is still very small ($J = 0.022$), suggesting that the information provided by the different ETs, derived from different sources, is in general complementary.

To investigate further, we create a sub-graph for each individual ET by preserving only edges of that type, and apply the four edge-only heuristics to evaluate on the SR task (test set) using only that sub-graph. SR is chosen because of its superior representativeness of companies (cf. question "⑧" in Appendix F.2) and alignment with our model selection approach (Section 4 and Appendix D). From Figure 3, we see some variation between the performance of different heuristics using different ETs. On many ETs, performance of all heuristics is close to random, since coverage is low (1%-47%, Figure 3), and few are able to yield performance much above chance. For ET9 (highest coverage), the full-path methods, USP and WSP, perform considerably better than immediate neighbors, but the same pattern is not seen, for example, on ET10 (also relatively high coverage). Nevertheless, the best performance is seen when all ETs are used together, giving full coverage of the companies in the samples, suggesting again the complementary nature of ETs. Here we see that full-path heuristics perform much better than immediate neighbors, reflecting useful structural information in the graph beyond the company-company relations that constitute the edges of the graph. Making use of the weights in WSP helps further, suggesting that they also carry useful information.

## 4.2 PERFORMANCE OF SP (SIMILARITY PREDICTION)

To evaluate the performance of the SP task, we compute the cosine similarity score between the embeddings of each pair of companies from all methods that produce vector embeddings of companies

| Approach / Model | | SP AUC | SR Acc% | CR Recall@50 | CR Recall@100 |
|---|---|---|---|---|---|
| Graph heuristics | USP (Unweighted SP) | N/A* | 59.61 | 18.27 | 29.48 |
| | WSP (Weighted SP) | N/A* | 61.16 | **43.69** | **56.03** |
| | Neighbors | 0.6229† | 56.52 | 22.25 | 31.84 |
| | W. Neighbors | 0.6020 | 56.65 | 43.50 | 54.65 |
| Embedding proximity | mSBERT (512-dim) | **0.8060** | 67.14 | 12.96 | 18.24 |
| | ADA2 (1536-dim) | 0.7450 | 67.20 | 14.09 | 21.69 |
| | SimCSE (768-dim) | 0.7188 | 61.69 | 7.66 | 8.90 |
| | PAUSE (32-dim) | 0.7542 | 65.19 | 6.84 | 9.62 |
| N-shot | ChatGPT-3.5 | 0.7501† | 66.73† | 30.06 | 31.10 |
| Knowledge Graph GNN-based Methods | GraphSAGE | 0.7422 | 62.90 | 10.12 | 11.93 |
| | (ADA2) | ±0.0202 | ±2.25 | ±3.91 | ±0.96 |
| | GRACE | 0.7243 | 59.36 | 2.68 | 4.64 |
| | (mSBERT) | ±0.0233 | ±0.64 | ±1.82 | ±1.51 |
| | MVGRL (PAUSE) | 0.6843 | 55.60 | 0.47 | 1.25 |
| | | ±0.0280 | ±1.18 | ±0.35 | ±0.56 |
| | GraphMAE | 0.7981 | **67.61** | 20.88 | 27.83 |
| | (mSBERT) | ±0.0063 | ±0.11 | ±0.46 | ±0.39 |
| | eGraphMAE | 0.7963 | 67.52 | 18.44 | 23.79 |
| | (mSBERT) | ±0.0030 | ±0.03 | ±0.21 | ±0.22 |

\* We omit SP evaluation for the path-based graph heuristics since, whilst they are able to rank companies by similarity, there is no obvious way to obtain a 0-1 similarity score for a pair of companies from them.

† To account for the binary nature of the SP prompts answered by ChatGPT and the Neighbors heuristic, we report accuracy instead of AUC.

Table 2: The performance of baselines on three evaluation tasks (SP, CR and SR) of CompanyKG. Best results are in bold. Standard deviations are reported over 3 random initializations of the trained models.

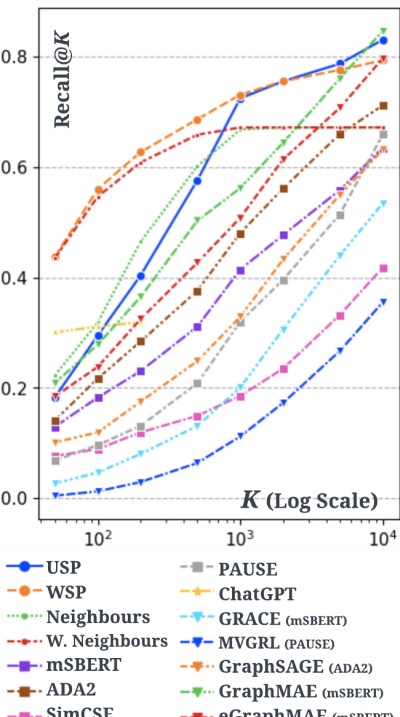

Figure 4: Performance comparison on CR task measured in Recall@$K$ over the full range of $K$ values tested.

(all except graph heuristics and ChatGPT). As the optimal threshold to distinguish between similar and dissimilar examples is unknown, we calculate the metric of Area Under the ROC Curve (AUC) for benchmarking, except for cases where the model can only produce a binary similar/dissimilar judgement (ChatGPT and Neighbors), where we report accuracy. We apply the immediate neighbor graph heuristics by classifying all pairs of companies that are directly connected by an edge in the graph as similar and others as dissimilar. We extend this to the weighted case by scaling weights to a 0-1 range and using them as similarity scores where edges exist.

Results are reported in Table 2. Edge-only baselines perform poorly: although the test data is similar in nature certain ETs in the graph, there is no overlap between the test pairs and the graph edges, so such a simple heuristic is not sufficient. Node-only baselines span a broad range, the best achieving ∼0.8, outperforming most other models. ChatGPT and GNN-based methods perform at a comparable level to the node-only baselines. Since we only display the results from the top-performing GNN models and across embedding types, chosen using the SR validation split, the de facto highest SP performance (0.8253), achieved by eGraphMAE+SimCSE, is slightly higher still than those shown here and is among the few methods that outperform mSBERT embedding proximity. (See Table 7 in Appendix.) As on other tasks, GraphMAE appears to be the most promising among the GNN-based methods, with a slight improvement brought by incorporating edge weights (eGraphMAE).

## 4.3 PERFORMANCE OF SR (SIMILARITY RANKING)

To evaluate SR task for KG and embedding proximity methods, we obtain embeddings for the query company and two candidate companies using each method. Next, we calculate cosine similarity scores between the query company and the two candidate companies resulting in two scores. The final prediction of the more similar company is made by choosing the candidate with the higher score. Evaluation using edge-only graph heuristics is described in Section 4.1.

We report the SR accuracy in Table 2. Edge-only methods establish a baseline of ∼60%, while some node-only methods (e.g., mSBERT and ADA2) are able to improve on this (∼65%), perhaps due to pretraining on extensive multilingual corpora. These baseline results suggest that a combination of node and edge information into a single representation might be able to take advantage of both sources of similarity prediction. However, the GNN-based methods struggle to improve on the baselines. The best node+edge method, GraphMAE, manages to slightly outperform the best node-only baseline,

achieving the overall best result. Incorporating edge weights into the model (eGraphMAE) does not succeed in improving on this result, even though we found using graph heuristics that the edge weights carry useful information.

## 4.4 PERFORMANCE OF CR (COMPETITOR RETRIEVAL)

For the CR task, we aim to determine how likely the direct competitors of a target company are to appear in the top-$K$ most similar companies, where $K$ represents the number of companies returned in the search results. To accomplish this, we perform an exhaustive search in the embedding space and return the top-$K$ companies with the highest cosine similarity scores to the target company. We then count the number of annotated direct competitors (denoted as $M$) that appear within the returned set and compute the metric of Recall@$K$, which is given by $\frac{M}{N}$, where $N$ is the total number of direct competitors in the CR test set. Although we test eight $K$ values over a wide range – 50, 100, 200, 500, 1,000, 2,000, 5,000 and 10,000 (cf. Table 7 in Appendix and the plot in Figure 4) – it is the lower end of this range that is of most interest for downstream applications: in any search/retrieval system, the user will only in practice be able to examine a small number of the top-ranked results. We therefore only report Recall@50 and @100 in Table 2. Nevertheless, the comparison of performance at higher $K$ in Figure 4 also provides further insight into the comparison, since the different types of models show different behaviours as $K$ increases.

Edge-only heuristic methods that incorporate edge weights, specifically WSP and W. Neighbors, clearly stand out as superior. This underscores the significance of both the graph structure and edge weights, as we saw with graph heuristics in Section 4.1. The generative graph learning methods, GraphMAE and eGraphMAE, yield robust results, comparable to USP and Neighbors, suggesting that they succeed in capturing some salient aspects of the graph structure, but not in exploiting the edge weights. However, other GNN-based methods like GRACE perform poorly across all $K$s in comparison to edge-only and most node-only baselines. Whilst the comparative performance of the methods is similar to the other tasks, the differences are greater in CR. This suggests CR is a challenging task and may benefit greatly from KG-based learning, making it suited for benchmarking a model's ability to harness node and edge simultaneously.

## 4.5 EMBEDDING SPACE VISUALIZATION

Intuitively, it is desirable for similar companies to cluster closely together in the embedding space, while dissimilar ones are preferably pulled apart. To gain qualitative insights from this perspective, we reduce the embeddings (from various baseline methods) to two dimensions using UMAP (McInnes et al., 2018), and color-code them by manually annotated sectors (cf. Figure 5e in Appendix B). We visualize the embeddings before and after GNN training in Appendix E.2 (Figure 7). It is noticeable that GNN-based methods tend to nudge companies within the same sector closer together, which could help with tasks like industry sector prediction. However, this improved embedding clustering has no clear positive correlation to the performance of the three evaluation tasks designed for similar company quantification.

## 5 CONCLUSION AND DISCUSSION

To represent and learn diverse company features and relations, we propose a large-scale heterogeneous graph dataset – CompanyKG, originating from a real-world investment platform. Specifically, 1.17 million companies are represented as nodes enriched with company description embeddings; and 15 different inter-company relations result in 51.06 million weighted edges. Additionally, we carefully compiled three evaluation tasks (SP, CR and SR), on which we present extensive benchmarking results for 11 reproducible baselines categorized into three groups: node-only, edge-only, and node+edge. While node-only methods show good performance in the SR task, edge-only methods clearly stand out as superior in addressing the CR task. On SP task, node-only and GNN-based approaches outperform edge-only ones by a large margin, with the best result obtained by eGraphMAE that attempts to incorporate node embeddings, edges and their weights. Although generative graph learning exhibits robust performance in general, no single method takes a clear lead across all CompanyKG tasks, calling for future work to develop more robust learning methods that can incorporate node and edge information effectively. Overall, CompanyKG can accelerate research around a major real-world problem in the investment domain – company similarity quantification – while also serving as a benchmark for assessing self-supervised graph learning methods. Given the continuous evolution of nodes and edges, a logical extension involves capturing KG snapshots over time, thereby transforming it into a spatio-temporal KG.

## ACKNOWLEDGMENTS

We are especially grateful to [HIDDEN FOR REVIEW] ...

## LIMITATIONS AND SOCIETAL IMPACTS

The known limitations include:

- Due to compliance and legal reasons, the true identity of individual companies is considered sensitive information. As detailed in Appendix F.2 ④, we only provide the company description embeddings (as node features) instead of raw textual descriptions from four different pretrained LMs. A comprehensive overview of the actions taken to prevent the recovery of true company identities is presented in Appendix F.2 ⑪.

- As discussed in Appendix F.2 ③, CompanyKG does not represent an exhaustive list of all possible edges due to incomplete information from various data sources. This leads to a sparse and occasionally skewed distribution of edges for individual ETs. Such characteristics pose additional challenges for downstream learning tasks.

The identified potential societal impacts are as follows:

- Privacy Concerns: The raw data sources we use to build CompanyKG contain proprietary information. As a result, we anonymize, aggregate and transform the raw data to prevent re-identification. See Appendix F.2 ④ and ⑪ for more details.

- Bias: The dataset might reflect or perpetuate existing biases in the business world. For example, companies from certain industries or regions might be under-represented. Appendix 3.2 presents a company profile analysis. Appendix F.2 ③ has a detailed overview about representativeness of included companies and relations.

- Legal and Regulatory Issues: The compliance experts from [ANONYMIZED] evaluated the dataset, data collection methods, and potential risks associated with the research. Their conclusion is that all published artifacts adhere to relevant laws, regulations, policies, and ethical guidelines. See Appendix F.3 ⑥.

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

## A  DATASET, SOURCE CODE AND LICENSE

The dataset, along with its machine-readable metadata, is hosted on CERN-backed Zenodo data repository: `https://zenodo.org/record/[ANONYMIZED]`. Its long-term maintenance is discussed in our datasheet detailed in Appendix F.

The source code is available on GitHub at `https://github.com/[ANONYMIZED]`. This includes reproducible code for the Analysis and Experiments in Sections 3 and 4 of the main paper, following the ML Reproducibility Checklist (Pineau et al., 2021). The GitHub repository also contains links to the dataset and information on how to cite.

The authors hereby state that they bear all responsibility in case of violation of rights, etc., and confirm that the data license is as follows:

- The large-scale heterogeneous graph, CompanyKG, is distributed under the license of Creative Commons Attribution-NonCommercial 4.0 International (CC BY-NC 4.0): `https://creativecommons.org/licenses/by-nc/4.0`.
- The source code is distributed under MIT license: `https://opensource.org/license/mit`.

## B  COMPANY (NODE) PROFILE

To aid in understanding node/company profiles, we collect five attributes (a snapshot as of February 2023) for each company. The attributes have different rate of missing values and are bucketed. The attributes include (1) **employee_size** (Figure 5a): the total number of employees; (2) **total_funding** (Figure 5b): the accumulative funding (in USD) received by the company; (3) **geo_region** (Figure 5c): the geographic region where the company is registered; (4) **duration_years** (Figure 5d): the number of years since the company was founded; (5) **sector** (Figure 1e): the top level of industry sectors of the company. As illustrated in Figure 5, the companies are distributed widely when inspected using each attribute, demonstrating the graph's sophistication and inclusiveness.

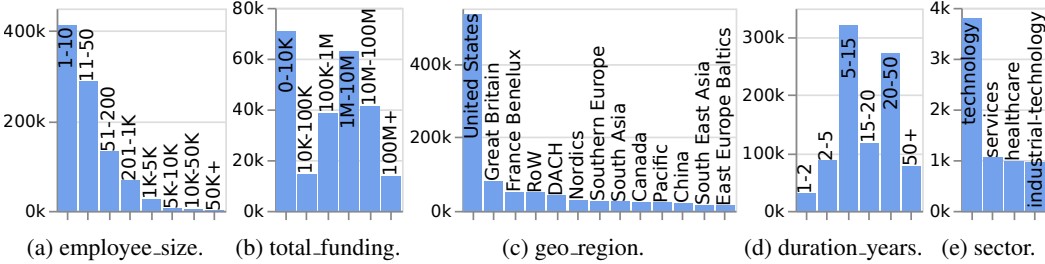

Figure 5: The distribution (y-axis) of bucketed company/node attributes (x-axis). In (c), "France Benelux" refers to France, Netherlands, Belgium and Luxembourg; "DACH" includes Germany, Austria and Switzerland; "RoW" stands for Rest of the World.

## C  RELATION (EDGE) WEIGHTS

To provide insight into the distribution of edge weights, we present histograms of the weights for all 15 ETs (ET1-15) in Figure 6. As shown in Table 1 in the main paper, edge weights are calculated differently depending on the nature of the edge/relation. Consequently, while some edge types (ET1-6 and ET11) have discrete weights, others follow a continuous distribution.

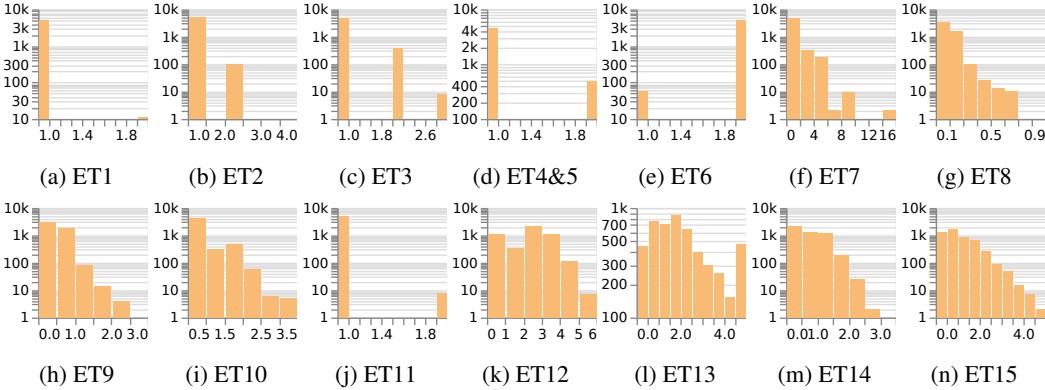

Figure 6: The weight distributions of different ETs: we randomly sample 5,000 edges for each type. The histograms of ET4 and ET5 are almost identical, hence they are merged.

## D  EXPERIMENTAL SETTINGS AND HYPER-PARAMETERS

In order to ensure the reproducibility and transparency of our experimental results, we provide a comprehensive overview of the experimental settings and hyper-parameter search strategy employed in our study. For each baseline method included in our benchmark, we present the specific hardware and software configuration used during the experiments. Additionally, we detail the range of hyper-parameters that were explored and the optimal hyper-parameters identified through the search process. By sharing this information, we aim to facilitate the replication of our experiments and promote a deeper understanding of the performance of the baseline methods in our evaluation.

### D.1  GRAPHSAGE

We carry out the GraphSAGE training and evaluation on a Linux Machine with 8 vCPUs, 52GB RAM, 1024GB SSD Disk, and one Nvidia Tesla P100 GPU. GraphSAGE is trained to generate beneficial embeddings for nodes, taking into account their local neighborhoods. It utilizes a mini-batch methodology in its training process, achieved by sampling a restricted number of neighboring nodes. Although this approach allows it to scale to large graphs, it also extends the time needed to complete a single epoch of training. As a result, we are limited to training GraphSAGE on CompanyKG dataset for a maximum of two epochs. We adopt the "SAGEConv" (Hamilton et al., 2017a) implementation in Deep Graph Library (DGL: https://www.dgl.ai) as the GNN layers, and we choose Graph

Convolutional Network (GCN) as the aggregator. The searched hyper-parameters for GraphSAGE are shown in Table 3.

| Hyper-parameter | Searched values | The selected value for different node features | | | |
|---|---|---|---|---|---|
| | | mSBERT | **ADA2** | SimCSE | PAUSE |
| The number of sampled neighbors | 6, 12 | 12 | **12** | 12 | 12 |
| The number of GNN layers | 2, 3 | 2 | **2** | 2 | 2 |
| Training batch size | $2^{10}, 2^{12}$ | $2^{12}$ | $\mathbf{2^{12}}$ | $2^{12}$ | $2^{12}$ |
| Training epochs | 1, 2 | 1 | **1** | 1 | 2 |
| Dropout rate | 0.1, 0.3 | 0.3 | **0.3** | 0.3 | 0.1 |
| Learning rate | $1 \times 10^{-3}, 1 \times 10^{-4}$ | $1 \times 10^{-4}$ | $\mathbf{1 \times 10^{-4}}$ | $1 \times 10^{-3}$ | $1 \times 10^{-3}$ |
| Embedding dimension | $2^5, 2^6, 2^7$ | $2^6$ | $\mathbf{2^7}$ | $2^7$ | $2^5$ |

Table 3: The researched and optimal hyper-parameters for GraphSAGE over different node features. The **bold column** corresponds to the optimal feature type and the main model reported in the results table of the main paper.

## D.2 GRACE AND MVGRL

GCL training with GRACE and MVGRL is run on a Linux machine with 8 vCPUs, 52GB RAM, 300GB SSD Disk, and one Nvidia V100 GPU. We train with a minibatch scheme that loads the neighbours of nodes sampled in each minibatch to a depth sufficient to compute the full loss function for the sampled nodes. Where nodes have many neighbors, we randomly subsample a fixed number to limit the memory required for a batch.

We use Adam optimization with a learning rate of 0.1, betas of 0.9 and 0.999 and no weight decay. In this case, we do not apply early stopping, but train every model for 100 epochs.

We run grid search over the remaining hyper-parameters: the number of subsampled neighbors, the number of layers of the GNN and the dimensionality of the learned hidden representations. The tested hyperparameter values are shown in Table 4 (GRACE) and 5 (MVGRL).

| Hyper-parameter | Searched values | Selected value for each node feature type | | | |
|---|---|---|---|---|---|
| | | **mSBERT** | ADA2 | SimCSE | PAUSE |
| The number of sampled neighbors | 5, 10 | **5** | 10 | 10 | 10 |
| The number of GNN layers | 2, 3 | **3** | 2 | 3 | 3 |
| Hidden layer dimensionality | 8, 16, 32 | **16** | 32 | 8 | 16 |

Table 4: The searched and optimal hyper-parameters for GRACE over different node feature types, selected based on performance on the SR validation set. The **bold column** corresponds to the optimal feature type and the main model reported in the results table of the main paper.

| Hyper-parameter | Searched values | Selected value for each node feature type | | | |
|---|---|---|---|---|---|
| | | mSBERT | ADA2 | SimCSE | **PAUSE** |
| The number of sampled neighbors | 5, 10 | 10 | 5 | 10 | **5** |
| The number of GNN layers | 2, 3 | 2 | 3 | 2 | **3** |
| Hidden layer dimensionality | 8, 16, 32 | 16 | 16 | 8 | **16** |

Table 5: The searched and optimal hyper-parameters for MVGRL over different node feature types, selected based on performance on the SR validation set. The **bold column** corresponds to the optimal feature type and the main model reported in the results table of the main paper.

## D.3 GRAPHMAE

GraphMAE is trained and evaluated on a Linux machine with 12 vCPUs, 85GB RAM, 1024GB SSD Disk, and one Nvidia Tesla A100 GPU. As GraphMAE is a full-batch algorithm and the entire graph cannot be accommodated in the GPU memory, we construct a randomized multi-scale mini-batch for each training step. Concretely, we randomly select an integer $p$ from

$\{5, 10, 15, 20, 30, 40, 50, 100, 300, 400, 600, 800, 1000\}$ and partition the graph into $p$ sub-graphs, following Chiang et al. (2019); then, we sample $\lceil * \rceil \frac{p}{10}$ sub-graphs to compose the current training mini-batch.

We train the model for a maximum of $1 \times 10^3$ epochs, implementing an early stop condition if the accuracy on the SR validation dataset does not improve for 100 epochs. We perform a grid search for the most important hyper-parameters, as shown in Table 6, while keeping the other hyper-parameters fixed to the default values selected by Hou et al. (2022): `alpha_l`=3, `replace_rate`=0.15, `loss_fn`="SCE", `encoder`=`decoder`="GAT", `norm`="LayerNorm", `num_out_heads`=1, `optimizer`="Adam", `pooling`="Mean", `residual`=True. During training, the learning rate $\ell$ is decayed at the end of each epoch using the following formula:

$$\ell_{m+1} = \frac{\ell_m}{2} \left[ 1 + \cos\left( \frac{m\pi}{1 \times 10^3} \right) \right], \tag{1}$$

where $\ell_m$ and $\ell_{m+1}$ denote the learning rate used in the current and next epoch, respectively.

| Hyper-parameter | Searched values | Selected value for each node feature type | | | |
| --- | --- | --- | --- | --- | --- |
| | | **mSBERT** | ADA2 | SimCSE | PAUSE |
| The number of GNN layers | 2, 3 | **2** | 2 | 2 | 2 |
| The number of attention heads | 4, 8 | **8** | 4 | 4 | 4 |
| Node mask rate | 0.1, 0.3, 0.5, 0.7 | **0.3** | 0.3 | 0.7 | 0.5 |
| Dropout rate* | 0.1, 0.3 | **0.3** | 0.1 | 0.1 | 0.1 |
| Learning rate | $1 \times 10^{-3}, 5 \times 10^{-4}$ | **$1 \times 10^{-3}$** | $5 \times 10^{-4}$ | $1 \times 10^{-3}$ | $1 \times 10^{-3}$ |
| Edge drop rate | 0.1, 0.3, 0.5, 0.7 | **0.1** | 0.5 | 0.5 | 0.3 |
| Embedding dimension | $2^6, 2^7, 2^8, 2^9$ | **$2^8$** | $2^9$ | $2^7$ | $2^7$ |

* It refers to the dropout rate of the attention dropout layer; and the dropout rate of the feature dropout layers is always set to 0.1 higher than the attention dropout rate up to a maximum of 1.0.

Table 6: The searched and optimal hyper-parameters for GraphMAE over different node feature types, selected based on performance on the SR validation set. The **bold column** corresponds to the optimal feature type and the main model reported in the results table of the main paper.

### D.4 EDGE-FEATURED GRAPHMAE (EGRAPHMAE)

We choose GAT (graph attention network) (Veličković et al., 2018) as encoder and decoder for GraphMAE, given its outstanding performance as reported by Hou et al. (2022). Within a GAT layer, the attention score $\alpha_{ij}$ between node $i$ and $j$ is calculated with

$$\alpha_{ij} = Softmax(LeakyReLU(\mathbf{a}^\top [\mathbf{W}\mathbf{h}_i || \mathbf{W}\mathbf{h}_j])) \tag{2}$$

where $\mathbf{h}_i$ and $\mathbf{h}_j$ are feature vectors of node $i$ and $j$, respectively; $\mathbf{W}$ is a learnable weight matrix; $\mathbf{a}^\top$ is the transpose of a learnable parameter vector; and notation "$||$" represents the vector concatenation operation. However, such an implementation, does not be utilize the edge weights available in graphs like CompanyKG. To that end, we replace GAT with EGAT (edge-featured GAT) (Wang et al., 2021), resulting in a new approach – eGraphMAE (edge-featured GraphMAE). Specifically, the attention score (between node $i$ and $j$) in a EGAT layer is calculated by

$$\alpha_{ij} = Softmax(\mathbf{v}^\top \boldsymbol{\epsilon}'_{ij}) \quad \text{and} \quad \boldsymbol{\epsilon}'_{ij} = LeakyReLU(\mathbf{A}[\mathbf{W}\mathbf{h}_i || \boldsymbol{\epsilon}_{ij} || \mathbf{W}\mathbf{h}_j]) \tag{3}$$

where $\mathbf{v}^\top$ is the transpose of a trainable parameter vector; $\mathbf{A}$ is a trainable parameter matrix; $\boldsymbol{\epsilon}_{ij}$ is the input embedding of the edge between node $i$ and $j$; and $\boldsymbol{\epsilon}'_{ij}$ is the output embedding to be used as the input for the next EGAT layer. It is important to note that the input edge embedding of the first EGAT layer is identical to the edge weight vectors provided in the graph, while the output edge embedding of the last EGAT layer is disregarded.

Since eGraphMAE requires significantly more memory than GraphMAE, we conduct the experiments for eGraphMAE on a Linux machine with 24 vCPUs, 170GB RAM, a 1024GB SSD Disk, and one Nvidia Tesla A100 GPU. Given the significant similarity between GraphMAE (Hou et al., 2022) and eGraphMAE (Appendix D.4), we adopt the same optimal set of hyper-parameter values as in GraphMAE (refer to Appendix D.3). However, in eGraphMAE, we introduce a new hyper-parameter for the number of dimensions in the output edge embeddings $\boldsymbol{\epsilon}'_{ij}$. We set this value to 32, which is the highest feasible value for our hardware configuration.

### D.5 CHATGPT: PROMPTING TEMPLATES FOR SP, SR AND CR TASKS

As one of our baselines, ChatGPT relies on the raw texts associated with the companies, which we are unable to include in the publicly released dataset due to legal and compliance constraints. Therefore, this baseline is incorporated solely for the sake of comparison.

**Prompting template for SP task**:

> *Below are the descriptions for two companies, company* A *and company* B. *Based on these descriptions, please indicate whether the companies are similar or not. The response should simply be "1" if they are similar and "0" otherwise, with no extra details.*
> *Company* A *description:* [...]
> *Company* B *description:* [...]
> *Response:* [TO_BE_FILLED_BY_LLM]

**Prompting template for SR task**: we also provide two prompts with ground truth labels as demonstrations.

> *You are a helpful and responsible assistant that help me to make a choice. Please answer as concisely as possible.*
> *Example:*
> *Given information of a query company and two candidate companies* $C_0$ *and* $C_1$. *Which candidate company (*$C_0$ *or* $C_1$*) is more similar to the target company?*
> *Query company description:* [...]
> *Company* $C_0$ *description:* [...]
> *Company* $C_1$ *description:* [...]
> *Your choice (Company* $C_0$ *or Company* $C_1$*) is:* $C_0$
> *Another example:*
> *Given information of a query company and two candidate companies* $C_0$ *and* $C_1$. *Which candidate company (*$C_0$ *or* $C_1$*) is more similar to the target company?*
> *Query company description:* [...]
> *Company* $C_0$ *description:* [...]
> *Company* $C_1$ *description:* [...]
> *Your choice (Company* $C_0$ *or Company* $C_1$*) is:* $C_1$
> *Now, get ready to answer the following questions:*
> *Given information of a query company and two candidate companies* $C_0$ *and* $C_1$. *Which candidate company (*$C_0$ *or* $C_1$*) is more similar to the target company?*
> *Query company description:* [...]
> *Company* $C_0$ *description:* [...]
> *Company* $C_1$ *description:* [...]
> *Your choice (Company* $C_0$ *or Company* $C_1$*) is:* [TO_BE_FILLED_BY_LLM]

**Prompting template for CR task**. ChatGPT performance on CR task varied significantly depending on the prompt formulation; and we used the prompt below to best mimic the evaluation of the other CR methods.

> *Below is the description of a company. Your task is to provide a unique list of* [$K$] *of its competitors, formatted as a python list and sorted from most similar to least similar. Make sure they are real companies.*
> *Target company description:* [...]
> *Competitor List:* [TO_BE_FILLED_BY_LLM]

The notation "[...]" denotes the textual information of involved companies, and "[$K$]" is a variable to be specified in Section 4.4 of the main paper.

## E EXTENDED EXPERIMENTAL RESULTS

Due to page limit, we can only present a selected subset of the experimental results in the main paper. We present the complete experimental results here.

| | Approach / Model | SP AUC | SR Acc% | CR Recall@50 | CR Recall@100 | CR Recall@200 | CR Recall@500 | CR Recall@1k | CR Recall@2k | CR Recall@5k | CR Recall@10k |
|---|---|---|---|---|---|---|---|---|---|---|---|
| Graph heuristics | USP | N/A* | 59.61 | 18.27 | 29.48 | 40.38 | 57.62 | 72.46 | 75.56 | **78.74** | 83.08 |
| | WSP (Weighted SP) | N/A* | 61.16 | **43.69** | **56.03** | **62.72** | **68.67** | **73.10** | **75.58** | 77.62 | 79.37 |
| | Neighbors | 0.6229$^\dagger$ | 56.52 | 22.25 | 31.84 | 46.50 | 60.16 | 66.81 | 67.19 | 67.19 | 67.19 |
| | W. Neighbors | 0.6020 | 56.65 | 43.50 | 54.65 | 60.86 | 65.86 | 67.19 | 67.19 | 67.19 | 67.19 |
| Embedding proximity | mSBERT (512-dim) | 0.8060 | 67.14 | 12.96 | 18.24 | 23.03 | 31.10 | 41.43 | 47.71 | 55.84 | 63.49 |
| | ADA2 (1536-dim) | 0.7450 | 67.20 | 14.09 | 21.69 | 28.41 | 37.61 | 48.05 | 56.13 | 66.00 | 71.18 |
| | SimCSE (768-dim) | 0.7188 | 61.69 | 7.66 | 8.90 | 11.94 | 14.93 | 18.55 | 23.48 | 33.21 | 41.79 |
| | PAUSE (32-dim) | 0.7542 | 65.19 | 6.84 | 9.62 | 13.11 | 20.84 | 31.96 | 39.60 | 51.45 | 65.92 |
| *N*-shot Prompt. | ChatGPT-3.5 | 0.7501$^\dagger$ | 66.73 | 30.06 | 31.10 | 31.91 | N/A$^\ddagger$ | N/A$^\ddagger$ | N/A$^\ddagger$ | N/A$^\ddagger$ | N/A$^\ddagger$ |
| mSBERT | GraphSAGE | 0.7415 | 62.03 | 10.80 | 13.04 | 17.63 | 25.29 | 33.68 | 43.29 | 53.82 | 64.30 |
| | | ±0.0102 | ±3.11 | ±3.61 | ±2.15 | ±1.60 | ±1.82 | ±0.82 | ±2.40 | ±1.11 | ±1.69 |
| | GRACE | 0.7243 | 59.36 | 2.68 | 4.64 | 8.03 | 13.10 | 20.03 | 30.61 | 43.98 | 53.46 |
| | | ±0.0233 | ±0.64 | ±1.82 | ±1.51 | ±1.34 | ±0.29 | ±0.28 | ±1.32 | ±2.21 | ±4.58 |
| | MVGRL | 0.7208 | 58.29 | 2.17 | 3.56 | 5.83 | 10.65 | 15.52 | 23.37 | 35.96 | 44.37 |
| | | ±0.0336 | ±0.74 | ±1.03 | ±0.33 | ±0.86 | ±1.25 | ±3.61 | ±2.79 | ±2.18 | ±0.47 |
| | GraphMAE | 0.7981 | **67.61** | 20.88 | 27.83 | 36.48 | 50.27 | 56.21 | 64.43 | 76.06 | 84.69 |
| | | ±0.0063 | ±0.11 | ±0.46 | ±0.39 | ±0.53 | ±0.26 | ±0.37 | ±0.76 | ±0.40 | ±0.40 |
| | eGraphMAE | 0.7963 | 67.52 | 18.44 | 23.79 | 32.47 | 42.68 | 50.82 | 61.39 | 70.73 | 79.69 |
| | | ±0.0030 | ±0.03 | ±0.21 | ±0.22 | ±0.20 | ±0.55 | ±0.41 | ±0.18 | ±0.88 | ±1.10 |
| ADA2 | GraphSAGE | 0.7422 | 62.90 | 10.12 | 11.93 | 17.51 | 24.85 | 32.87 | 43.26 | 54.98 | 63.13 |
| | | ±0.0202 | ±2.25 | ±3.91 | ±0.96 | ±2.44 | ±0.92 | ±1.90 | ±1.74 | ±1.92 | ±1.42 |
| | GRACE | 0.7548 | 58.20 | 3.09 | 4.83 | 7.60 | 12.88 | 21.40 | 31.63 | 44.33 | 54.71 |
| | | ±0.0264 | ±0.47 | ±0.59 | ±0.78 | ±1.20 | ±2.79 | ±2.45 | ±5.07 | ±5.58 | ±4.24 |
| | MVGRL | 0.6638 | 55.85 | 1.00 | 1.77 | 3.27 | 7.11 | 10.85 | 15.42 | 24.04 | 33.38 |
| | | ±0.0557 | ±0.88 | ±0.35 | ±0.24 | ±0.90 | ±1.56 | ±3.33 | ±3.11 | ±6.59 | ±7.50 |
| | GraphMAE | 0.8132 | 65.10 | 24.14 | 29.15 | 35.06 | 46.24 | 58.21 | 67.53 | 75.61 | **88.32** |
| | | ±0.0053 | ±0.06 | ±0.30 | ±0.26 | ±0.36 | ±0.52 | ±0.67 | ±0.36 | ±0.21 | ±1.20 |
| | eGraphMAE | OOM! | OOM! | OOM! | OOM! | OOM! | OOM! | OOM! | OOM! | OOM! | OOM! |
| SimCSE | GraphSAGE | 0.7390 | 59.90 | 10.26 | 14.07 | 17.47 | 25.71 | 38.13 | 45.84 | 54.80 | 64.77 |
| | | ±0.0095 | ±2.26 | ±0.92 | ±3.06 | ±1.24 | ±0.86 | ±2.25 | ±1.87 | ±0.70 | ±0.94 |
| | GRACE | 0.7149 | 57.26 | 0.39 | 1.07 | 2.25 | 4.34 | 6.93 | 11.77 | 19.25 | 29.53 |
| | | ±0.0524 | ±1.38 | ±0.57 | ±0.86 | ±1.65 | ±3.31 | ±4.87 | ±6.69 | ±9.51 | ±11.09 |
| | MVGRL | 0.7180 | 55.82 | 0.79 | 1.20 | 2.14 | 4.95 | 8.36 | 14.11 | 21.22 | 29.33 |
| | | ±0.0600 | ±0.69 | ±0.10 | ±0.45 | ±1.03 | ±1.57 | ±2.31 | ±3.77 | ±4.06 | ±4.68 |
| | GraphMAE | 0.8108 | 65.46 | 19.67 | 26.61 | 33.24 | 44.32 | 53.06 | 63.45 | 76.45 | 84.39 |
| | | ±0.0066 | ±0.05 | ±1.14 | ±0.63 | ±2.20 | ±1.16 | ±2.32 | ±2.67 | ±1.77 | ±0.95 |
| | eGraphMAE | **0.8253** | 66.09 | 18.53 | 26.33 | 33.42 | 44.39 | 51.50 | 63.71 | 74.69 | 83.81 |
| | | ±0.0089 | ±0.38 | ±0.19 | ±0.17 | ±0.91 | ±0.58 | ±1.02 | ±0.76 | ±0.52 | ±0.24 |
| PAUSE | GraphSAGE | 0.7421 | 60.09 | 5.79 | 8.02 | 12.20 | 17.22 | 24.82 | 31.98 | 42.84 | 55.22 |
| | | ±0.0121 | ±1.83 | ±1.24 | ±2.03 | ±1.77 | ±1.81 | ±1.70 | ±0.89 | ±7.07 | ±2.47 |
| | GRACE | 0.6698 | 58.15 | 0.66 | 1.31 | 2.07 | 5.44 | 9.58 | 15.40 | 26.27 | 36.47 |
| | | ±0.0135 | ±0.78 | ±0.36 | ±0.36 | ±0.48 | ±0.71 | ±0.80 | ±1.51 | ±4.46 | ±3.97 |
| | MVGRL | 0.6843 | 55.60 | 0.47 | 1.25 | 2.94 | 6.39 | 11.19 | 17.29 | 26.68 | 35.52 |
| | | ±0.0280 | ±1.18 | ±0.35 | ±0.56 | ±0.77 | ±1.76 | ±2.67 | ±2.72 | ±0.79 | ±1.83 |
| | GraphMAE | 0.7727 | 64.81 | 10.16 | 12.51 | 14.87 | 19.59 | 25.41 | 31.90 | 42.85 | 54.33 |
| | | ±0.0113 | ±0.40 | ±1.13 | ±0.62 | ±0.43 | ±0.46 | ±1.32 | ±0.12 | ±1.13 | ±2.75 |
| | eGraphMAE | 0.7742 | 65.17 | 11.62 | 13.95 | 15.42 | 19.28 | 26.88 | 37.49 | 49.92 | 62.53 |
| | | ±0.0068 | ±0.95 | ±0.33 | ±1.50 | ±0.97 | ±0.44 | ±0.53 | ±1.28 | ±0.75 | ±1.47 |

The leftmost grouping label for the GNN-based rows reads vertically: "Knowledge Graph (initiated with different types of node embeddings) GNN-based Methods".

\* We omit SP evaluation for the path-based graph heuristics, since, whilst they are able to rank companies by similarity, there is no obvious way to obtain a 0-1 similarity score for a pair of companies from them.

$\dagger$ To account for the binary nature of the SP prompts answered by ChatGPT, we report accuracy (Acc.) instead of AUC.

$\ddagger$ As ChatGPT's answer on the CR task is not limited to the companies in CompanyKG and is not mapped to any specific company/node IDs, we conducted a manual examination of the top-$K$ responses and counted the number of entries that match the ground truth competitors. As a result, when the value of $K$ exceeds 200, it becomes overly tedious to calculate hit rate for ChatGPT.

Table 7: The performance of the popular and state-of-the-art baselines on three evaluation tasks of CompanyKG: SP, SR and CR which are compared with Area Under the ROC Curve (AUC), Accuracy (Acc.) and Recall@$K$ respectively. Best results are in bold. "OOM!" means out-of-memory.

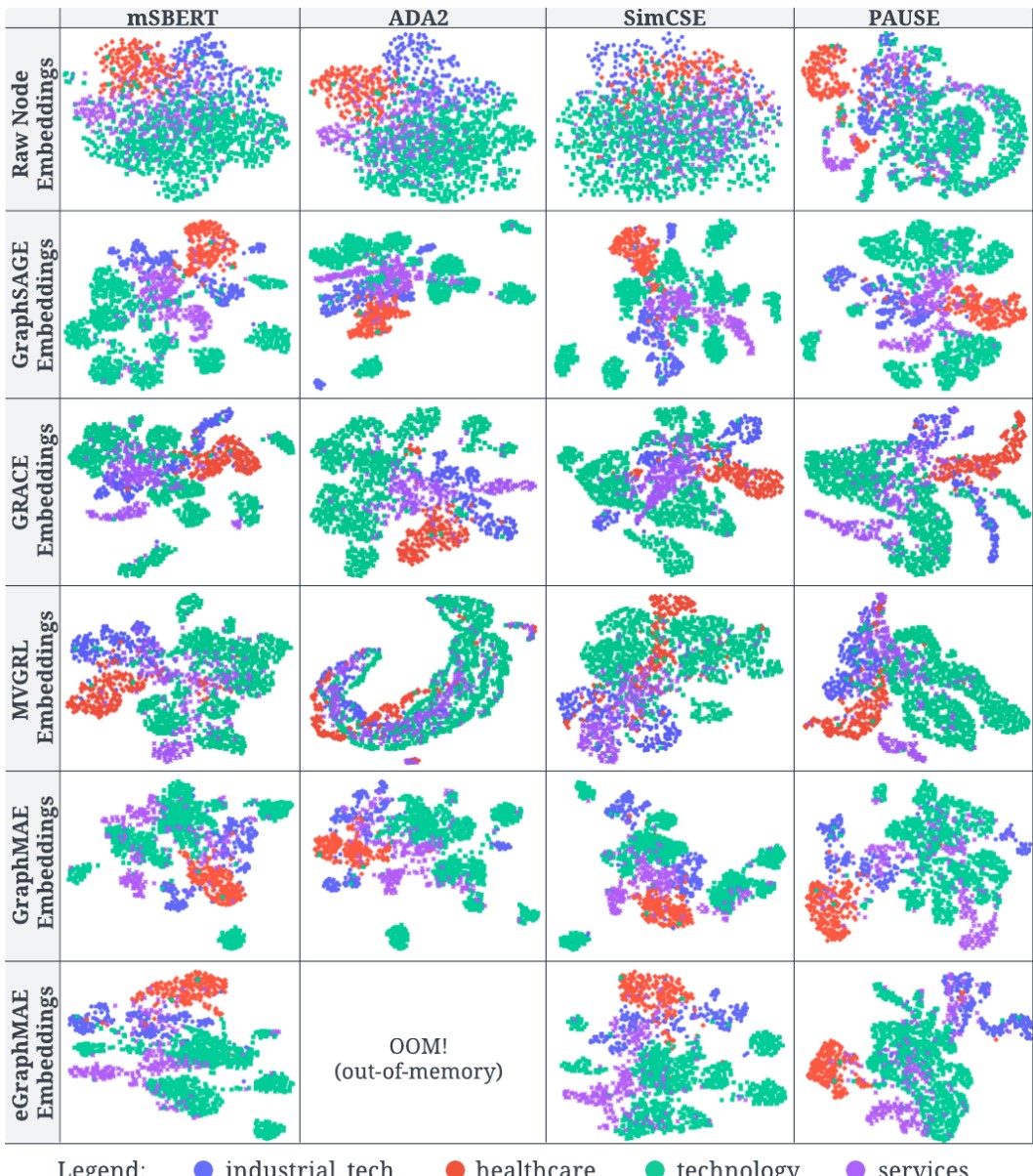

Figure 7: Dimension reduction and visualization of embeddings obtained from different LMs (the first row) and GNN-based models (the rest of the rows).

### E.1 Performance of SP, CR and SR

In Section 4.2, 4.3 and 4.4 of the main paper, we treat the type of node features as a hyper-parameter and only report the model that performs the best on SR validation split. In Table 7, we report the SP, SR and CR performance of each individual model trained using different node feature (mSBERT, ADA2, SimCSE and PAUSE). It is worth noting that the best performing models on SP and CR (Recall@10k) are different than the results (Table 2) in the main paper.

### E.2 Embedding Space Visualization

Figure 7 is the visualization for Section 4.5 in the main paper. We reduce the embeddings (from either node feature or various GNN-based baselines) to two dimensions using UMAP (McInnes et al., 2018), and color-code them by manually annotated sectors (cf. Figure 5e in Appendix B).

# F    DATASHEET OF COMPANYKG

This Datasheet follows the template defined by Gebru et al. (2021).

## F.1    MOTIVATION

**①  For what purpose was the dataset created?** Was there a specific task in mind? Was there a specific gap that needed to be filled? Please provide a description.

Please refer to Section 1 in the main paper for the answer to this question.

**②  Who created the dataset (e.g., which team, research group) and on behalf of which entity (e.g., company, institution, organization)?**

This dataset was created by [HIDDEN FOR REVIEW] ...

**③  Who funded the creation of the dataset?** If there is an associated grant, please provide the name of the grantor and the grant name and number.

The development of this dataset was financially supported by [HIDDEN FOR REVIEW], covering expenses related to human resources and access to proprietary and third-party data sources as mentioned in Table 1 of the main paper.

## F.2    COMPOSITION

**①  What do the instances that comprise the dataset represent (e.g., documents, photos, people, countries)?** Are there multiple types of instances (e.g., movies, users, and ratings; people and interactions between them; nodes and edges)? Please provide a description.

CompanyKG is a heterogeneous graph consisting of nodes and undirected edges, with each node representing a real-world company and each edge signifying a relationship between the connected pair of companies. As illustrated in Figure 5, the nodes/companies span a diverse range across five distinct attributes: (1) the total number of employees; (2) the cumulative funding received by the company; (3) the geographic region where the company is registered; (4) the number of years since the company was founded; and (5) the industry sectors of the company. The graph edges capture 15 different inter-company relationships, as outlined in Table 1 in the main paper, resulting in 15 unique edge types (abbreviated as ET), i.e., ET1∼ET15.

CompanyKG also includes three evaluation datasets (summarized below) representing three different tasks related to similar company quantification. Section 3.3 provides more details.

- Similarity Prediction (SP): contains pairs of companies that are either annotated as similar (denoted by "1") or dissimilar (denoted by "0");
- Similarity Ranking (SR): each sample consists of one query company and two candidate companies (identified by 0 and 1, respectively), and its label (valued 0 or 1) indicates which candidate is more similar to the query company.
- Competitor Retrieval (CR): contains several target companies, each of which is annotated with $N$ ($N \geq 1$) direct competitors by investment professionals.

**②  How many instances are there in total (of each type, if appropriate)?**

CompanyKG comprises a total of 1,169,931 nodes and 50,815,503 edges; the distribution of edges across each edge type is summarized in Table 1 of the main paper. Regarding evaluation tasks, the SP dataset contains 3,219 annotated company pairs; the SR dataset includes 1,856 company ranking questions with ground-truth answers; and the CR dataset comprises 76 target companies, each with ∼5.3 manually identified direct competitors in average.

**③  Does the dataset contain all possible instances or is it a sample (not necessarily random) of instances from a larger set?** If the dataset is a sample, then what is the larger set? Is the sample representative of the larger set (e.g., geographic coverage)? If so, please describe how this representativeness was validated/verified. If it is not representative of the larger set, please describe why not (e.g., to cover a more diverse range of instances, because instances were withheld or unavailable).

The inclusion criteria is aligned with [ANONYMIZED]'s investment focus concerning aspects such as geographic location, industry sector, total funding, time of establishment and number of employees. An insight into the coverage of companies from those aspects can be found in Appendix B. That said, CompanyKG includes only a subset of companies still operating as of the publication date of this dataset.

Certain edge types, such as ET8 (employee flow), ET9 (shared keywords), ET11 (mergers and/or acquisitions), ET12 (event co-attendance), ET13 (common potential customers), ET14 (co-appearance in news), and ET15 (common executives) in Table 1 of the main paper, reflect more universal relations and are likely to lead to a more representative subset of companies. In contrast, other relations (e.g., ET1 to ET3) tend to result in a skewed subset of companies, as they often represent specific interests and focuses of the investment professionals.

Acknowledging the incompleteness and skewness of edges from individual relation types, we merge all relations into a single heterogeneous (in terms of edge types) large graph, expecting the resulting company set to better represent the entire population of companies. Appendix B displays the distribution of the included companies from five perspectives, demonstrating their wide distribution across each attribute, which highlights the graph's complexity and inclusiveness.

④ **What data does each instance consist of?** "Raw" data (e.g., unprocessed text or images) or features? In either case, please provide a description.

We compile a textual description for each company (corresponding to a node in CompanyKG) by extracting relevant information (i.e., company name, keywords and descriptions) from multiple data sources. Due to legal and compliance concerns, we cannot disclose the complete list of these sources. For example, the assembled textual description for company "Klarna" should look like

> *Klarna is a fintech company that provides support for direct and post-purchase payments. Keywords: "buy-now-pay-later", "shopping", "saving account" and "financial service".*

It is worth noting that about 5% company descriptions are in languages other than English. Since the true identity of individual companies is considered sensitive information, we can only publish the company description embeddings (a.k.a. node features) from four different pretrained LMs. The choice of LMs encompasses both pretrained off-the-shelf models (mSBERT and ADA2) and proprietary fine-tuned models (SimCSE and PAUSE). We hereby justify our selection of LMs and offer additional details about the chosen ones below.

- mSBERT (multilingual Sentence BERT) (Reimers & Gurevych, 2019) is selected due to its early and wide adoption in generating contextually rich and language-aware sentence embeddings, catering to the multilingual nature of our dataset. The model is downloaded from `https://huggingface.co/sentence-transformers/distiluse-base-multilingual-cased-v2`.

- ADA2 (i.e., "text-embedding-ada-002" model introduced in `https://platform.openai.com/docs/guides/embeddings`) is GPT-3's most recent embedding engine employing a 1536-dimensional semantic space, with default settings used throughout the experiments. At the time of writing this paper, ADA2 represents the most comprehensive and state-of-the-art language embedding model publicly accessible, serving as a versatile and general-purpose solution to encode the description of companies.

- SimCSE (Gao et al., 2021) address the reality of label scarcity (i.e., company pairs that are annotated as similar or dissimilar) well, as it obviates the need for annotations during the fine-tuning process of this embedding model. Specifically, we finetune a "bert-base-ucased" model (Devlin et al., 2019) in a self-supervised manner using all company descriptions for two epochs.

- PAUSE, a semi-supervised language embedding model, displays the ability to minimize approximately 90% of the normal labeling effort while maintaining superior embedding performance. Its practical application spans over two years in various investment scenarios, including market mapping and M&A. For this work, we adopt the "PAUSE-SC-10%" model introduced in Section 4.4 of (Cao et al., 2021).

Readers can refer to Figure 1 (top-right part) of the main paper for a specific example of node features or embeddings.

**⑤ Is there a label or target associated with each instance?** If so, please provide a description.

No. But we have three evaluation datasets (cf. Section 3.3 in main paper) with ground-truth labels, which represent different subsets of all instances.

**⑥ Is any information missing from individual instances?** If so, please provide a description, explaining why this information is missing (e.g., because it was unavailable). This does not include intentionally removed information, but might include, e.g., redacted text.

While we have gathered company descriptions from multiple data sources, there are still instances (¡0.2%) where the description is either too brief or too generic to contain sufficient relevant information useful for measuring inter-company similarity. In these cases, the discriminative ability of the node embeddings may be somewhat diminished.

Regarding the edges incorporated into CompanyKG, they do not constitute a comprehensive representation of all conceivable edges, primarily due to incomplete information from various data sources. Such incompleteness may be attributed to the data source selection process, which mostly aligns with [ANONYMIZED]'s investment preferences, or to the quality of the chosen data sources.

**⑦ Are relationships between individual instances made explicit (e.g., users' movie ratings, social network links)?** If so, please describe how these relationships are made explicit.

The undirected and weighted edges in CompanyKG represent 15 distinct inter-company relations, as detailed in Section 3.1 and Table 1 of the main paper, leading to 15 unique edge types (abbreviated as ET), specifically ET1∼ET15. Specifically, for any two companies that share at least one type of relationship, we create a single undirected edge between them with a 15-dimensional weight vector (refer to the bottom-right portion of Figure 1 in the main paper). The $i$-th value of the weight vector is the weight of the $i$-th edge type (i.e., ET$i$).

**⑧ Are there recommended data splits (e.g., training, development/validation, testing)?** If so, please provide a description of these splits, explaining the rationale behind them.

In reflecting real use cases within the PE industry, the primary goal of constructing CompanyKG is to develop similar company search algorithms using the entire graph. The performance of these algorithms should be measured based on three evaluation tasks – SP, SR, and CR, as described in Section 3.3 of the main paper. Consequently, there is no recommended graph split.

However, the graph can be naturally divided into 15 sub-graphs by retaining only one edge type at a time, as demonstrated in the empirical analyses conducted in Sections 4.1 of the main paper.

We further randomly split the SR evaluation dataset into validation and test sets at a 1:5 ratio. It is recommended to use the SR validation set (comprising 368 samples/questions) for selecting the optimal combination of hyper-parameters and to report the selected model on the SR test set and the entire SP and CR evaluation datasets. The rationale for not splitting the SP or CR datasets further is two-fold:

(1) The SP and CR datasets used for validation are constructed using output from historical [ANONYMIZED]'s investment tasks, leading to a potential bias towards the areas of interest for [ANONYMIZED]'s investment professionals. As a result, the generalizability of the SP and CR tasks may be limited.

(2) The binary similarity labels used in the SP task do not capture the fine-grained similarity discrimination required by many real-world use cases. Excelling at the SP task does not necessarily indicate the ability to differentiate between companies with subtle differences in similarity. Additionally, the SP task requires the use of a uniform threshold to distinguish between similar and dissimilar cases, which may not align with the varying similarity thresholds in different scenarios.

**⑨ Are there any errors, sources of noise, or redundancies in the dataset?** If so, please provide a description.

Each relation that we model as an edge type in the graph has a varying degree of incompleteness. In most cases, quantifying the degree of completeness is challenging, especially for edge types that rely on third-party information, as explained in Table 1 of the main paper. Consequently, we recommend using all edge types together when training graph learning algorithms on CompanyKG to mitigate this issue.

Furthermore, both node features and edge weights can be prone to noise:

- The node features are based on company description embeddings, and the quality of these embeddings is influenced by the completeness and relevance of the raw textual descriptions.

- The statistics used to calculate edge weights may be somewhat inaccurate. For instance, inaccuracies may arise in determining people's affiliations and durations when deriving edge weights for ET8.

Lastly, considering that CompanyKG integrates multiple data sources, we established an entity resolution system to merge information from different data sources. During that merging process, two types of errors may emerge: (1) incorrectly merging two different companies into a single graph node and (2) representing the same company as different nodes.

**(10) Is the dataset self-contained, or does it link to or otherwise rely on external resources (e.g., websites, tweets, other datasets)?** If it links to or relies on external resources, a) are there guarantees that they will exist, and remain constant, over time; b) are there official archival versions of the complete dataset (i.e., including the external resources as they existed at the time the dataset was created); c) are there any restrictions (e.g., licenses, fees) associated with any of the external resources that might apply to a dataset user? Please provide descriptions of all external resources and any restrictions associated with them, as well as links or other access points, as appropriate.

The dataset is self-contained and does not rely on or link to external resources. During the construction of CompanyKG, some third-party data sources were used to build a portion of the edges and nodes. However, all information has been incorporated by performing encoding, transformation, and anonymization to meet [ANONYMIZED]'s compliance requirements. For more details, please refer to Appendix F.3.

**(11) Does the dataset contain data that might be considered confidential (e.g., data that is protected by legal privilege or by doctor–patient confidentiality, data that includes the content of individuals' non-public communications)?** If so, please provide a description.

There are primarily three categories of information considered confidential and/or proprietary:

- The disclosure of individual companies' true identities may risk leaking deal information due to certain inter-company relations derived from ET1, 2, 3, 7, and 10. As a result, we conceal the names of the companies and use incremental numerical IDs to identify them.

- The company descriptions are assembled using several third-party data sources, so we cannot directly publish them. Additionally, publishing the company descriptions (even with the company names hidden) could potentially enable speculation about the true identities of some companies. For instance, it would be relatively easy to guess the company "King" from the description "COMPANY_NAME *is a mobile game developer and publisher that gained prominence after releasing Candy Crush Saga in 2012*". To address this issue, we convert the company descriptions into NLP embeddings (a.k.a. node features) using four different pretrained LMs, as already introduced in F.2. To prevent reverse engineering of the embeddings (to recover company identities), we also apply linear transformations to the NLP embeddings, selecting different transformation parameters for each embedding type. The details of this transformation are not shared.

- The details of the data sources, whether from [ANONYMIZED] or third-party, used to construct the 15 different edge types are considered as sensitive proprietary information. Nevertheless, we have made our best effort to explain each edge type in Table 1 of the main paper. The rationale behind this is that disclosing the exact third-party data sources could potentially enable the identification of the true identities of some companies by comparing their connection patterns using criteria such as in/out-degree, centrality, and other relevant graph connectivity measures.

**(12) Does the dataset contain data that, if viewed directly, might be offensive, insulting, threatening, or might otherwise cause anxiety?** If so, please describe why.

No, the dataset focuses on companies and their relationships, with all sensitive information being concealed or anonymized as previously discussed. Consequently, it does not contain data that, if viewed directly, might be offensive, insulting, threatening, or cause anxiety.

⑬ **Does the dataset relate to people?** If not, you may skip the remaining questions in this section.

No, this dataset focuses on companies and their relationships with one another. Although some relationships, such as ET8 and ET15, are built upon people's affiliations, they are heavily aggregated, making it impossible to identify any individual person.

## F.3    COLLECTION PROCESS

① **How was the data associated with each instance acquired?** Was the data directly observable (e.g., raw text, movie ratings), reported by subjects (e.g., survey responses), or indirectly inferred/derived from other data (e.g., part-of-speech tags, model-based guesses for age or language)? If the data was reported by subjects or indirectly inferred/derived from other data, was the data validated/verified? If so, please describe how.

The data used to construct CompanyKG primarily originates from two categories of sources:

- First-hand, high-quality information from the [ANONYMIZED] investment platform, which includes historical user interaction records. This data assists in constructing edge types ET1, 2, 3, 7, and 10, as presented in Table 1 of the main paper. The information is considered reliable because it is directly provided by [ANONYMIZED]'s investment professionals to facilitate various investment activities.

- Third-party data sources offer information for constructing the remaining edge types and raw textual company descriptions. All data providers in this category have signed commercial contracts with us, which include quality assurance measures as a crucial aspect.

As previously introduced in Appendix F.2, the published node features (refer to Section 3.2 of the main paper) and edge definitions and weights (refer to Section 3.1 of the main paper) are not directly observable due to required anonymization, embedding, and transformation processes. This ensures the data is in compliance with privacy and confidentiality requirements while maintaining its utility for research purposes.

② **What mechanisms or procedures were used to collect the data (e.g., hardware apparatus or sensor, manual human curation, software program, software API)?** How were these mechanisms or procedures validated?

As previously emphasized, CompanyKG is built upon the integration of company information from various internal and external data sources. The information about companies continually arrives (from the [ANONYMIZED] platform or external data providers) into our streaming data pipeline, which primarily performs two consecutive tasks:

(1) Associating the incoming company information with a company ID using an entity resolution system as mentioned in the response to question "⑨" in Appendix F.2.

(2) Materializing the updated company entity with the newly associated information in our data warehouse, as explained in a blog post titled [HIDDEN FOR REVIEW].

CompanyKG is constructed using the materialized company entities as of April 5th, 2023.

Collection of annotated test sets for the three evaluation tasks is described in Section 3.3 the main paper. The annotation procedure for the SR task is slightly more involved, so warrants some more detail here.

Ten [ANONYMIZED] employees initially labeled 119 samples. Subsequently, around 15 experienced annotators labeled 2,400 carefully curated samples (including the initial set) through the annotation service Appen (https://appen.com). The task is a choice between two candidate companies, the order of which is randomized prior to annotation. Multiple rounds of annotation were carried out in order to ensure that the professional annotators' understanding of the task was consistent with those of the [ANONYMIZED] employees, and in the process some annotators were excluded. The final set annotated by Appen annotators was combined with the initial set from [ANONYMIZED] employees. During Appen annotation, three annotations from different annotators were collected per sample first. In cases without full agreement (1,280/2,281), an annotation was collected from a fourth annotator. 544 samples that still did not reach a majority agreement (i.e. had 2/4 votes for each candidate) were excluded from the final set, leaving 1,856. The final label set exhibits a slight bias towards label 0

(56% vs 44%), which we attribute to a tendency by some annotators to select the first candidate when uncertain. In a supervised setting, where a model is trained on a portion of the annotated set, we would therefore consider 56% accuracy to be baseline performance. However, in the experiments reported in the main paper, none of the models has access to this dataset at training time and almost all models outperform this hypothetical baseline.

**③ If the dataset is a sample from a larger set, what was the sampling strategy (e.g., deterministic, probabilistic with specific sampling probabilities)?**

We direct readers to our response to question "③" in Appendix F.2 for a detailed explanation on this matter.

**④ Who was involved in the data collection process (e.g., students, crowdworkers, contractors) and how were they compensated (e.g., how much were crowdworkers paid)?**

The integration of multi-source information into unified [ANONYMIZED] company entities is performed by the Data Engineers of [ANONYMIZED]. The construction of the CompanyKG dataset from the materialized company entities is carried out by the authors of this paper, listed in the response to question "②" in Appendix F.1. The persons involved in creating the three evaluation datasets (SP, SR, and CR) are described below.

- SP: Two [ANONYMIZED] employees were involved in manually verifying the labels.
- SR: Ten [ANONYMIZED] employees (mainly data scientists and machine learning engineers) were involved in labeling the first 119 questions. Subsequently, around 15 experienced labelers hired by Appen (`https://appen.com`) labeled 2,400 carefully curated questions at a total cost of approximately 18,000 EUR. More details of this process are given under question "②" above.
- CR: Four experienced [ANONYMIZED] employees manually extracted the data from PE deal materials.

**⑤ Over what timeframe was the data collected?** Does this timeframe match the creation timeframe of the data associated with the instances (e.g., recent crawl of old news articles)? If not, please describe the timeframe in which the data associated with the instances was created.

The CompanyKG dataset (including the graph and evaluation tasks) represents a snapshot of the [ANONYMIZED] platform's data warehouse as of April 5th, 2023. The authors began working on incrementally building and iterating the dataset in September 2022. However, the materialized company entities have been integrating information from various data sources since 2016.

**⑥ Were any ethical review processes conducted (e.g., by an institutional review board)?** If so, please provide a description of these review processes, including the outcomes, as well as a link or other access point to any supporting documentation.

Yes, the compliance experts from [ANONYMIZED] evaluated the dataset, data collection methods, and potential risks associated with the research. Their conclusion is that all published artifacts adhere to relevant laws, regulations, policies, and ethical guidelines.

### F.4 PREPROCESSING/CLEANING/LABELING

**① Was any preprocessing/cleaning/labeling of the data done (e.g., discretization or bucketing, tokenization, part-of-speech tagging, SIFT feature extraction, removal of instances, processing of missing values)?** If so, please provide a description. If not, you may skip the remaining questions in this section.

Firstly, we encode the textual company description into four different NLP embeddings, as detailed in the response to question "①" in F.2. The published node embeddings are also linearly transformed to prevent recovering the true company identities, as described in the answer to question "⑪" in F.3.

Secondly, we aggregate the relation (i.e., edge) strength into a single scalar, which is truncated, normalized, or log-transformed as specified in Table 1 of the main paper. As introduced in Section 3.1 of the main paper, we merge multiple edges/connections (if any) between the same pair of companies into one by assigning a 15-dimensional edge weight vector. It is crucial to note that a value of "0" in any edge weights vector represents "unknown relation" rather than "no relation", which can be regarded as a form of edge imputation.

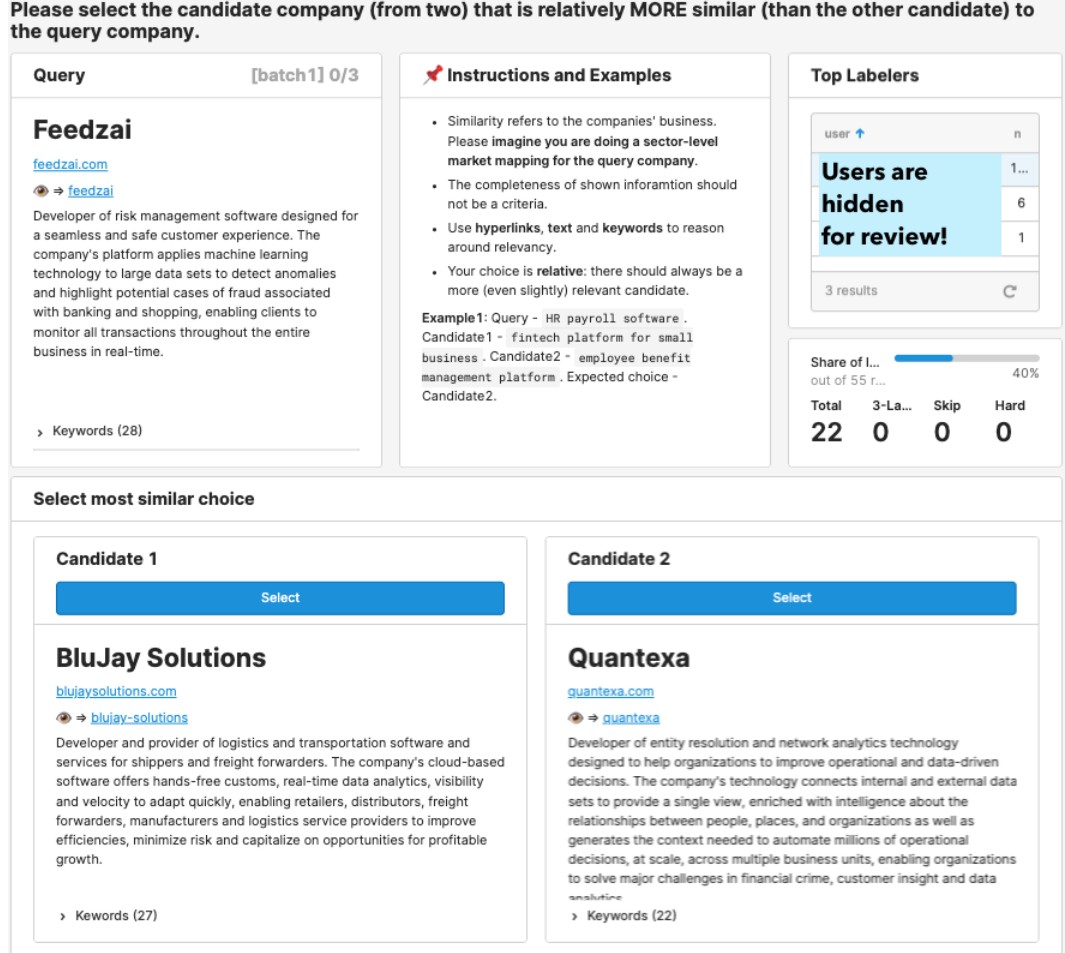

Figure 8: A screenshot of the web application developed for labeling SR (Similarity Ranking) task.

②  **Was the "raw" data saved in addition to the preprocessed/cleaned/labeled data (e.g., to support unanticipated future uses)?** If so, please provide a link or other access point to the "raw" data.

No, due to compliance requirements discussed earlier, the "raw" data cannot be publicly shared.

③  **Is the software that was used to preprocess/clean/label the data available?** If so, please provide a link or other access point.

Given that the "raw" data isn't publicly available, we do not release the software for preprocessing or cleaning it. For the SP task, [ANONYMIZED]'s proprietary platform, [ANONYMIZED], is utilized to acquire labels. The CR task, which involves manually extracting direct competitor information from meticulously chosen investment deep-dive documents, does not necessitate custom software. Conversely, for the SR task, we developed a labeling web application using Retool (https://retool.com), as depicted in Figure 8. This tool was initially designed to validate the labeling experience and quality. Subsequently, Appen (https://appen.com) incorporated it into their platform for labeling the complete SR dataset.

## F.5   USES

①  **Has the dataset been used for any tasks already?** If so, please provide a description.

No, although the non-anonymized dataset is planned to be used internally for applications like market/competitor mapping and M&A.

②**Is there a repository that links to any or all papers or systems that use the dataset?** If so, please provide a link or other access point.

The CompanyKG dataset has been archived on Zenodo (https://zenodo.org), complete with its own DOI for easy referencing. This allows for tracking of all papers that cite this dataset. At present, we do not have plans to maintain a separate manual repository for tracking dataset usage. However, we remain open to reconsidering this stance if we receive substantial requests.

③**What (other) tasks could the dataset be used for?**

CompanyKG serves as a robust benchmark for any unsupervised graph learning algorithms aimed at generating node embeddings. The meticulously designed evaluation tasks within our dataset represent high-quality, manually curated benchmarks that provide multi-faceted assessments of the learned company embeddings. These tasks can significantly aid in comparing and evaluating the performance of various embedding techniques.

④**Is there anything about the composition of the dataset or the way it was collected and preprocessed/cleaned/labeled that might impact future uses?** For example, is there anything that a dataset consumer might need to know to avoid uses that could result in unfair treatment of individuals or groups (e.g., stereotyping, quality of service issues) or other risks or harms (e.g., legal risks, financial harms)? If so, please provide a description. Is there anything a dataset consumer could do to mitigate these risks or harms?

As outlined in Appendix F.4, the "raw" data undergoes a series of processes including encoding, aggregation, transformation, and anonymization to ensure proprietary information remains secure. Consequently, the dataset is not suitable for analyzing any real-world investment scenarios.

⑤**Are there tasks for which the dataset should not be used?** If so, please provide a description.

It is imperative that users of this dataset refrain from attempting to decipher the true identities of the companies contained within.

### F.6 DISTRIBUTION

①**Will the dataset be distributed to third parties outside of the entity (e.g., company, institution, organization) on behalf of which the dataset was created?** If so, please provide a description.

Yes, we have designated this dataset as open access to ensure its widest possible utilization. In an effort to promote reuse and potential contributions, we have deliberately opted for the most permissive licenses available for this dataset.

②**How will the dataset be distributed (e.g., tarball on website, API, GitHub)?** Does the dataset have a digital object identifier (DOI)?

Yes, the dataset DOI is [ANONYMIZED]. Please see Appendix A.

③**When will the dataset be distributed?**

After the review process of the conference.

④**Will the dataset be distributed under a copyright or other intellectual property (IP) license, and/or under applicable terms of use (ToU)?** If so, please describe this license and/or ToU, and provide a link or other access point to, or otherwise reproduce, any relevant licensing terms or ToU, as well as any fees associated with these restrictions.

See Appendix A.

⑤**Have any third parties imposed IP-based or other restrictions on the data associated with the instances?** If so, please describe these restrictions, and provide a link or other access point to, or otherwise reproduce, any relevant licensing terms, as well as any fees associated with these restrictions.

None.

⑥**Do any export controls or other regulatory restrictions apply to the dataset or to individual instances?** If so, please describe these restrictions, and provide a link or other access point to, or otherwise reproduce, any supporting documentation.

None.

### F.7 MAINTENANCE

**①Who will be supporting/hosting/maintaining the dataset?**

The authors from [ANONYMIZED], akin to typical academic practice, will ensure the long-term upkeep of the dataset's content. In relation to hosting, we have entrusted Zenodo with this responsibility to ensure maximal availability and longevity of the dataset. Backed by CERN, Zenodo has become the gold standard for dataset distribution, offering excellent availability and redundancy. More information about the underlying infrastructure and redundancy measures of Zenodo can be found at `https://about.zenodo.org/infrastructure`.

**②How can the owner/curator/manager of the dataset be contacted (e.g., email address)?**

The primary contacts are [HIDDEN FOR REVIEW] ...

**③Is there an erratum?** If so, please provide a link or other access point.

No.

**④Will the dataset be updated (e.g., to correct labeling errors, add new instances, delete instances)?** If so, please describe how often, by whom, and how updates will be communicated to dataset consumers (e.g., mailing list, GitHub)?

All amendments to the dataset, including error corrections and expansions concerning the number of nodes, edges, and evaluation samples, will be handled through Zenodo. Each updated version will be assigned a unique DOI by Zenodo, while all versions collectively will be identifiable under a fixed root DOI. Our commitment is to maintain the dataset's quality and growth, responding to any errors that are identified and planning for future enhancements.

**⑤If the dataset relates to people, are there applicable limits on the retention of the data associated with the instances (e.g., were the individuals in question told that their data would be retained for a fixed period of time and then deleted)?** If so, please describe these limits and explain how they will be enforced.

No retention limits.

**⑥Will older versions of the dataset continue to be supported/hosted/maintained?** If so, please describe how. If not, please describe how its obsolescence will be communicated to dataset consumers.

As the dataset is hosted on Zenodo, which supports DOI versioning, all different versions of the dataset are appropriately tracked and stored. This versioning functions much like incremental updates, duplicating only those files that have undergone modification. For more detailed information on this process, please refer to Zenodo's versioning guidelines at `https://help.zenodo.org/#versioning`.

**⑦If others want to extend/augment/build on/contribute to the dataset, is there a mechanism for them to do so?** If so, please provide a description. Will these contributions be validated/verified? If so, please describe how. If not, why not? Is there a process for communicating/distributing these contributions to dataset consumers? If so, please provide a description.

We encourage users to share additional benchmarking results using various graph learning algorithms. Furthermore, we welcome discussions regarding potential relationships that could expand the graph. Please refer to question "②" above for contact details. Any suggestions will be thoroughly evaluated and validated internally on a case-by-case basis.

