# OpenReview forum: "CompanyKG: A Large-Scale Heterogeneous Graph for Company Similarity Quantification"
_ICLR.cc/2024/Conference — ICLR 2024 Conference Withdrawn Submission_

### Official Review · Reviewer_3BPH · 2023-10-27

**Soundness:** 3 good
**Presentation:** 3 good
**Contribution:** 2 fair
**Rating:** 3
**Confidence:** 3

**Summary:**

The paper introduces a new large knowledge graph dataset called CompanyKG for quantifying company similarity, defines relevant benchmarking tasks, and reports comprehensive results of different baseline methods on the tasks. It provides a new resource for developing graph-based approaches for applications in investment and related domains. However, the contribution of the paper was limited to the company investment research area and the results don't seem to have broader influence to the ICLR research community. Thus, I am not inclined to accept this paper at the current stage.

**Strengths:**

1. The paper is well-written. It clearly explains the motivation, background, dataset construction process, task definitions, and benchmarking methodology.
2. The paper constructed a novel CompanyKG dataset, incorporating over 1 million companies and 50 million relationships, which is one of the first public available dataset in this field.
3. Extensive experiments on various tasks and methods were provided as the benchmark.

**Weaknesses:**

1. the contribution of the paper was limited to the company investment research area and the results don't seem to have broader influence to the ICLR research community.

**Questions:**

1. Could you provide some case studies for the proposed tasks? It would be more intuitive for readers to understand this dataset and its value.

---

### Official Review · Reviewer_cqXj · 2023-10-30

**Soundness:** 2 fair
**Presentation:** 3 good
**Contribution:** 2 fair
**Rating:** 3
**Confidence:** 4

**Summary:**

This paper introduces CompanyKG, a large-scale heterogeneous graph tailored for quantifying company similarity. It encompasses 1.17 million companies, characterized by 15 distinct inter-company relations, amounting to 51.06 million weighted edges. To facilitate a holistic evaluation of company similarity quantification approaches, the authors present three assessment tasks accompanied by annotated test sets.

**Strengths:**

1.	The paper offers a comprehensive overview of CompanyKG, a large-scale heterogeneous graph crafted for measuring company similarity. It details its size, architecture, and associated evaluation tasks. The data presented is both substantial and pertinent.
2.	To ensure a thorough evaluation of company similarity quantification techniques, the paper incorporates three assessment tasks, each with its own annotated test set. This approach promises to advance further research in the field.

**Weaknesses:**

1.	The rationale behind using competitor companies as benchmarks for assessing investment potential is ambiguous. While the authors suggest that these companies aid in evaluation, it's unclear how they contribute to discerning investment potential. Is this truly a pivotal consideration in making investment decisions?
2.	The dataset's applicability to broader scenarios remains uncertain. The study primarily focuses on tasks associated with similar quantification, neglecting other essential knowledge graph tasks like relation extraction among companies and predicting a company's industry.
3.	The methodology for constructing the test dataset is nebulous. There's no explicit mention of the criteria used to select companies for this dataset. Additionally, the annotation procedure lacks clarity. What guidelines did the annotators follow? An in-depth analysis, including consistency checks, would substantiate the dataset's annotation accuracy.
4.	Although the creation of the company knowledge graph is touted as a significant contribution of the paper, the GNN-based methods designed for the Knowledge Graph don't outperform the embedding proximity method in SP and SR tasks. This raises questions about the unique value the constructed knowledge graph brings to the table, especially in the context of company similarity quantification.
5.	The section on related works could benefit from a more comprehensive coverage. It notably omits in-depth discussions on state-of-the-art techniques such as GNN-based approaches on Knowledge Graphs (KG), including KBGAT [1] and RGHAT [2], as well as Language Model (LM) based methods on KGs like SimKGC [3] and LMKE [4]. Additionally, the article lacks baseline comparisons for these methodologies.

**Reference**

[1] Deepak Nathani, Jatin Chauhan, Charu Sharma, and Manohar Kaul. 2019. Learning Attention-based Embeddings for Relation Prediction in Knowledge Graphs. In Proceedings of the 57th Annual Meeting of the Association for Computational Linguistics, pages 4710–4723, Florence, Italy. Association for Computational Linguistics.

[2] Zhang, Z., Zhuang, F., Zhu, H., Shi, Z., Xiong, H. and He, Q. 2020. Relational Graph Neural Network with Hierarchical Attention for Knowledge Graph Completion. Proceedings of the AAAI Conference on Artificial Intelligence. 34, 05 (Apr. 2020), 9612-9619.

[3] Liang Wang, Wei Zhao, Zhuoyu Wei, and Jingming Liu. 2022. SimKGC: Simple Contrastive Knowledge Graph Completion with Pre-trained Language Models. In Proceedings of the 60th Annual Meeting of the Association for Computational Linguistics (Volume 1: Long Papers), pages 4281–4294, Dublin, Ireland. Association for Computational Linguistics.

[4] Wang, X.; He, Q.; Liang, J.; and Xiao, Y. 2022. Language models as knowledge embeddings. In Proceedings of the Thirty-First International Joint Conference on Artificial Intelligence, IJCAI-22

**Questions:**

The related works in section 2.1 have discussed the text embedding proximity of LLaMA and GPT-3/4 and some works on GPT embeddings have proved its effectiveness such as [5]. However, in the experiments, the Large Language Model embedding method was not discussed in the text embedding proximity method. An explanation is needed here.

---

### Official Review · Reviewer_tTAV · 2023-10-31

**Soundness:** 2 fair
**Presentation:** 2 fair
**Contribution:** 2 fair
**Rating:** 3
**Confidence:** 4

**Summary:**

This paper presents a dataset CompanyKG and evaluates it on three tasks: similarity prediction, competitor retrieval and similarity ranking, over a set of existing methods.

**Strengths:**

1.	The dataset of company information is expected to be used by other research studies if it is released publicly.
2.	The three tasks evaluated over the datasets are useful in real applications.
3.	The experiments are conducted over various methods.

**Weaknesses:**

1.	Releasing one dataset may not be sufficient for the contribution of a paper.
2.	In title, it said heterogeneous graph, while in the paper, it is called knowledge graph. Though these two terms are related, but I think they are different, instead of using in a mixed way.
3.	The presentation of the paper can be improved. There is no need of so many places in bold.
4.	In experiments, the evaluation on the three tasks in Sec 4.2,4.3,4.4 is quite brief, with only one experiment per task. To thoroughly evaluate the performance of existing methods on these tasks, the experiments should be more comprehensive

**Questions:**

Please see the weaknesses.

---

### Official Review · Reviewer_Fmgf · 2023-11-02

**Soundness:** 2 fair
**Presentation:** 4 excellent
**Contribution:** 2 fair
**Rating:** 3
**Confidence:** 4

**Summary:**

The paper presents a novel dataset called CompanyKG, which has been meticulously developed to provide a knowledge graph that captures relationships between companies in real-world. The key contributions are:
1. CompanyKG Dataset: A comprehensive knowledge graph that encodes information about companies and their technological relationships.
2. Benchmarking: The paper provides baseline performances on several downstream tasks using CompanyKG, offering a foundation for future research to build upon.

**Strengths:**

S1: The development of CompanyKG constitutes a considerable undertaking in the realm of data engineering, illustrating a commendable dedication to aggregating and organizing an extensive array of nodes and edges—sourced from authentic real-world data—into a well-integrated and coherent knowledge graph. \
S2: The manuscript exhibits a commendable level of transparency in confronting the intrinsic limitations, possible biases, and societal implications tied to CompanyKG. This transparency is intrinsic to the integrity of the work, engendering trust and establishing precise expectations for the knowledge graph's deployment. The forthrightness in these disclosures is vital, allowing users to judiciously assess the utility of CompanyKG and to apply it with due responsibility.

**Weaknesses:**

W1: The paper's primary contribution appears to lean more towards advancements in data engineering and potential applications in financial research rather than pushing the envelope in AI theoretical development or algorithmic innovation. While the creation of CompanyKG is a notable achievement, the research may have limited implications for the direct progression of AI methodologies, as it focuses predominantly on the assembly and structuring of a large-scale knowledge graph. Consequently, this work might be seen as peripheral to core AI research interests, which traditionally emphasize novel approaches to learning, reasoning, or decision-making in artificial systems. \
W2: A significant weakness of CompanyKG is its static composition in the fast-changing corporate world. The paper does not detail methods for updating the knowledge graph to reflect new data, which is critical for maintaining its relevance and utility. \
W3: The third weakness of the paper is the absence of novel models or theoretical contributions, focusing primarily on the compilation and presentation of the knowledge graph without advancing AI research methodologies. \
W4: The paper does not delineate the update mechanisms or the associated costs for maintaining the contemporaneity of CompanyKG, a critical consideration given the dynamic nature of company data.

**Questions:**

Q1: How frequently and comprehensively is CompanyKG updated, and what are the associated costs of these updates?

**Details Of Ethics Concerns:**

In the final sections of the paper, the authors discussed privacy concerns and regulatory issues.